# Unlocking Point Processes through Point Set Diffusion

**David Lüdke,**[*] **Enric Rabasseda Raventós,**[*] **Marcel Kollovieh,** **Stephan Günnemann**
Department of Informatics & Munich Data Science Institute
Technical University of Munich, Germany
`{d.luedke,e.rabasseda,m.kollovieh,s.guennemann}@tum.de`

## Abstract

Point processes model the distribution of random point sets in mathematical spaces, such as spatial and temporal domains, with applications in fields like seismology, neuroscience, and economics. Existing statistical and machine learning models for point processes are predominantly constrained by their reliance on the characteristic intensity function, introducing an inherent trade-off between efficiency and flexibility. In this paper, we introduce POINT SET DIFFUSION, a diffusion-based latent variable model that can represent arbitrary point processes on general metric spaces without relying on the intensity function. By directly learning to stochastically interpolate between noise and data point sets, our approach effectively captures the distribution of point processes and enables efficient, parallel sampling and flexible generation for complex conditional tasks. Experiments on synthetic and real-world datasets demonstrate that POINT SET DIFFUSION achieves state-of-the-art performance in unconditional and conditional generation of spatial and spatiotemporal point processes while providing up to orders of magnitude faster sampling.

## 1 Introduction

Point processes describe the distribution of point sets in a mathematical space where the location and number of points are random. On Euclidean spaces, point processes (e.g., spatial and/or temporal; SPP, STPP, TPP) have been widely used to model events and entities in space and time, such as earthquakes, neural activity, transactions, and social media posts.

Point processes can exhibit complex interactions between points, leading to correlations that are hard to capture effectively (Daley & Vere-Jones, 2007). The distribution of points is typically characterized by a non-negative intensity function, representing the expected number of events in a bounded region of space (Daley et al., 2003). A common approach to modeling point processes on general metric spaces is to parameterize an inhomogeneous intensity as a function of space. However, this approach assumes independence between points, which restricts its ability to model complex interactions and hinders generalization across different point sets (Daley et al., 2003; Daley & Vere-Jones, 2007).

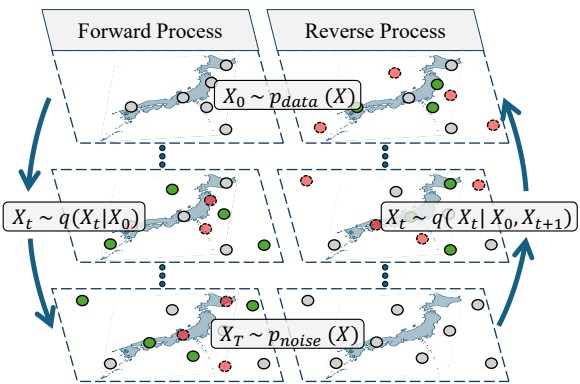

Figure 1: Illustration of POINT SET DIFFUSION for earthquakes in Japan. The forward process stochastically interpolates between the original data point set $X_0$ and a noise point set $X_T$, progressively removing data points and adding noise points. To generate new samples from the data distribution, we approximate the reverse posterior $q(X_t|X_0, X_{t+1})$ and add approximate data points and remove noise points.

---

[*]Equal contribution
Code is available at `https://www.cs.cit.tum.de/daml/point-set-diffusion`

For ordered spaces like time (STPP, TPP), the predominant approach is to model the conditional intensity autoregressively, conditioning each point on its past, allowing temporal causal dependencies, which can be conveniently captured by state-of-the-art machine learning models (Shchur et al., 2021). While this enables point interactions, these models rely on likelihood-based training and sequential sampling, which require integrating the intensity function over the entire space. Ultimately, this constrains models, as it either necessitates oversimplified parameterizations that restrict point dependencies and introduce smoothness (Ozaki, 1979; Ogata, 1998; Zhou & Yu, 2023), or approximations using amortized inference (Zhou et al., 2022), numerical (Chen et al., 2020), or Monte Carlo methods (Hong & Shelton, 2022). Thus, capturing complex point dependencies and sampling from point processes, particularly on general metric spaces, remains an open research challenge.

Lüdke et al. (2023) overcame the limitations of the conditional intensity function for temporal point processes by proposing ADD-THIN, a diffusion model for TPPs based on the thinning and superposition property of TPPs directly modeling entire event sequences. In this paper, we generalize this idea to point processes on general metric spaces and derive a diffusion-based latent variable model, POINT SET DIFFUSION, that directly learns to model the stochastic interpolation between a data point set and samples from any noise point process (see Figure 1). Furthermore, we show how to generate conditional samples with our unconditional POINT SET DIFFUSION model to solve arbitrary conditioning tasks on general metric spaces. Our experiments demonstrate that POINT SET DIFFUSION achieves state-of-the-art results on conditional and unconditional tasks for SPPs, TPPs and STPPs. Our contributions can be summarized as follows:

- We derive a diffusion-based latent variable model that captures the complex distribution of point processes on general metric spaces by learning stochastic interpolations between data and noise point sets.
- Our model enables efficient and parallel sampling of point sets while supporting flexible conditioning through binary functions on the metric space.
- We propose a model-agnostic evaluation framework for assessing generative point process models on Euclidean spaces.
- Our method achieves state-of-the-art results for conditional and unconditional generation of SPPs, TPPs, and STPPs while offering orders of magnitude faster sampling.

## 2 BACKGROUND

### 2.1 POINT PROCESSES

A *point process* (Daley et al., 2003) is a stochastic process where realizations consist of finite sets of points randomly located in a mathematical space. More formally, let $(D, d)$ be a complete, separable metric space equipped with its Borel $\sigma$-algebra $\mathcal{B}$. A point process on $D$ is a mapping $X$ from a probability space $(\Omega, \mathcal{A}, \mathcal{P})$ into $N^{lf}$, the set of all possible point configurations, such that for any bounded Borel set $A \subseteq D$, the number of points in $A$, denoted by $N(A)$, is a finite random variable.

Given a realization of the point process $X = \{\boldsymbol{x}_i \in D\}_{1 \leq i \leq n}$, where $n$ is the number of points, the number of points in a region is expressed as the *counting measure* $N(A) = \sum_{i=1}^{n} \mathbf{1}\{\boldsymbol{x}_i \in A\}$. Here, we assume the point process is simple, i.e., almost surely $N(\{\boldsymbol{x}_i\}) \leq 1$ for all $\boldsymbol{x}_i \in D$, meaning no two points coincide. Point processes are commonly characterized by their *intensity function*, which is defined through the following random measure:

$$A \mapsto \mu(A) := \mathbb{E}[N(A)] = \int_A \lambda(\boldsymbol{x}) \, \mathrm{d}\boldsymbol{x}, \tag{1}$$

where $\mu(A)$ represents the expected number of points in a region $A$. Then, a point process is said to have intensity $\lambda$ if the measure $\mu$ above has a *density* $\lambda$ with respect to the Lebesgue measure $\mu(A)$. Thus, the intensity function $\lambda(\boldsymbol{x})$ gives the expected number of points per unit volume in a small region of the Borel set $A \subseteq D$.

The points in a realization $X$ can exhibit complex correlations, so the intensity function is non-trivial to parameterize. On a Euclidean space $\mathbb{R}$ we can specify the Papangelou intensity (Daley et al., 2003):

$$\lambda(\boldsymbol{x}) = \lim_{\delta \to 0} \frac{P\{N(B_\delta(\boldsymbol{x})) = 1 | C(N(\mathbb{R} \setminus B_\delta(\boldsymbol{x})))\}}{|B_\delta(\boldsymbol{x})|}, \tag{2}$$

where $B_\delta(\boldsymbol{x})$ is the ball centered at $x$ with a radius of $\delta$, and $C(N(\mathbb{R} \setminus B_\delta(\boldsymbol{x})))$ represents the information about the point process outside the ball. If the Euclidean space is ordered, for instance, representing time, the conditioning term would represent the history of all points prior to $x$.

In general, effectively modeling and sampling from the *conditional intensity* (or related measures, e.g., hazard function or conditional density), for arbitrary metric spaces is a fundamental problem (Daley et al., 2003; Daley & Vere-Jones, 2007). This difficulty has led to a variety of simplified parametrizations that restrict the captured point interactions (Ozaki, 1979; Zhou & Yu, 2023; Daley et al., 2003; Daley & Vere-Jones, 2007); discretizations of the space (Ogata, 1998; Osama et al., 2019); and numerical or Monte Carlo approximations (Chen et al., 2020; Hong & Shelton, 2022).

In contrast, we propose a method that bypasses the abstract concept of a (conditional) intensity function by directly manipulating point sets through a latent variable model. Our approach leverages the following point process properties:[1]

*Superposition:* Given two point processes $N_1$ and $N_2$ with intensities $\lambda_1$ and $\lambda_2$ respectively, we define the superposition of the point processes as $N = N_1 + N_2$ or equivalently $X_1 \bigcup X_2$. Then, the resulting point process $N$ has intensity $\lambda = \lambda_1 + \lambda_2$. *Independent thinning:* Given a point process $N$ with intensity $\lambda$, randomly removing each point with probability $p$ is equivalent to sampling points from a point process with intensity $(1-p)\lambda$.

## 2.2 DIFFUSION MODELS

Ho et al. (2020) and Sohl-Dickstein et al. (2015) introduced a new class of generative latent variable models – probabilistic denoising diffusion models. Conceptually, these models learn to reverse a probabilistic nosing process to generate new data and consist of three main components: a *noising process*, a *denoising process*, and a *sampling procedure*. The *noising process* is defined as a forward Markov chain $q(X_{t+1}|X_t)$, which progressively noises a data sample $X_0 \sim p_{\text{data}}(X)$ over $T$ steps, eventually transforming it into a sample from a stationary noise distribution $X_T \sim p_{\text{noise}}(X)$. Then, the *denoising process* is learned to reverse the noising process by approximating the posterior $q(X_t|X_0, X_{t+1})$ with a model $p_\theta(X_t|X_{t+1})$. Finally, the *sampling procedure* shows how to generate samples from the learned data distribution $p_\theta(X) = \int p_{\text{noise}}(X_T) \prod_{t=0}^{T-1} p_\theta(X_t|X_{t+1}) \, \mathrm{d}X_1 \ldots \mathrm{d}X_T$.

# 3 POINT SET DIFFUSION

In this section, we derive a diffusion-based latent variable model for point sets on general metric spaces by systematically applying the thinning and superposition properties of random sets. This approach allows direct manipulation of random point sets, avoiding the need for the abstract concept of an intensity function. We begin by outlining the forward noising process in Section 3.1, which stochastically interpolates between point sets from the generating process and those from a noise distribution. Subsequently, we demonstrate how to learn to reverse this noising process to generate new random point sets in Section 3.2. Finally, in Section 3.3, we show how to sample from our unconditional model and generate conditional samples for general conditioning tasks on the metric space.

## 3.1 FORWARD PROCESS

Let $X_0 \sim p_{\text{data}}(X)$ be an i.i.d. sample from the generating point process, and let $X_T \sim p_{\text{noise}}(X)$ represent a sample from a noise point process. We define the forward process as a stochastic interpolation between the point sets $X_0$ and $X_T$ over $T$ steps. This process is modeled as a Markov chain $q(X_{t+1}|X_t)$, where $X_t$ is the superposition of two random subsets: $X_t^{\text{thin}} \subseteq X_0$ and $X_t^\epsilon \subseteq X_T$. Specifically, $\forall t : X_t = X_t^{\text{thin}} \bigcup X_t^\epsilon$, where $X_t^{\text{thin}}$ and $X_t^\epsilon$ are independent samples from a *thinning* and a *noise* process, respectively. We define the *thinning* and *noise* processes given two noise schedules $\{\alpha_t \in (0,1)\}_{t=1}^T$ and $\{\beta_t \in (0,1)\}_{t=1}^T$ as follows:

**Thinning Process:** This process progressively thins points in $X_0^{\text{thin}} = X_0$, removing signal over time. At every step $t+1$, each point $\boldsymbol{x} \in X_t^{\text{thin}}$ is independently thinned with probability $1 - \alpha_{t+1}$:

$$q(\boldsymbol{x} \in X_{t+1}^{\text{thin}} | \boldsymbol{x} \in X_t^{\text{thin}}) = \alpha_{t+1}. \tag{3}$$

---

[1]We provide a proof of both properties for general Borel sets in A.1.

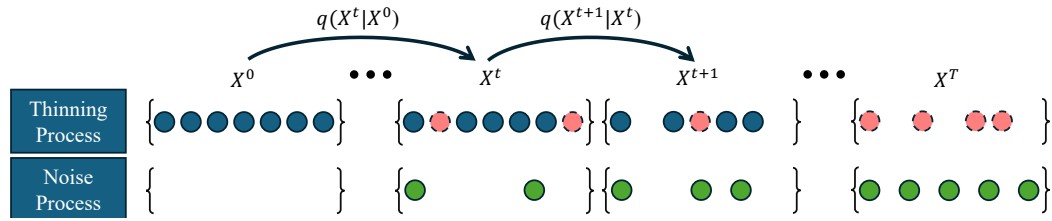

Figure 2: The forward process is a Markov Chain $q(X_{t+1}|X_t)$, that stochastically interpolates a data sample $X_0$ with a noise point set $X_T$ over $T$ steps by applying a *thinning* and a *noise* process.

Consequently, the thinning defines $n$ independent Bernoulli chains, and the probability of any point $\boldsymbol{x} \in X_0$ remaining in $X_t^{\text{thin}}$ is:

$$q(\boldsymbol{x} \in X_t^{\text{thin}}|\boldsymbol{x} \in X_0) = \bar{\alpha}_t, \tag{4}$$

where $\bar{\alpha}_t = \prod_{i=1}^n \alpha_i$. Equivalently, the intensity of the thinned points at step $t$ is given by $\lambda_t^{\text{thin}} = \bar{\alpha}_t \lambda_0$ and the number of remaining points follows a Binomial distribution: $Pr[n_t|X_0^{\text{thin}}] = \text{Binomial}(|X_0|, \bar{\alpha}_t)$, where $n_t = |X_t^{\text{thin}}|$.

**Noise Process:** This process adds random points $X_T^\epsilon \sim p_{\text{noise}}(X)$ sampled from a noise point process with intensity $\lambda^\epsilon$. At step $t+1$, we express $X_{t+1}^\epsilon|X_t^\epsilon$ as:

$$X_{t+1}^\epsilon = X_t^\epsilon \cup X_{t+1}^{\Delta\epsilon}, \quad \text{where } X_{t+1}^{\Delta\epsilon} \sim \beta_{t+1}\lambda^\epsilon. \tag{5}$$

By the superposition property, the intensity of $X_t^\epsilon$ is $\lambda_t^\epsilon = \bar{\beta}_t \lambda^\epsilon$, where $\bar{\beta}_t = \sum_{i=1}^t \beta_i$ and $\bar{\beta}_t \in [0,1]$. Alternatively, we can view the noise process as a reversed thinning process: we sample $X_T^\epsilon \sim p_{\text{noise}}(X)$ and thin it by $1 - \bar{\beta}_t$ to obtain $X_t^\epsilon$. Given a noise sample $X_T^\epsilon$, we then find that:

$$q(\boldsymbol{x} \in X_t^\epsilon|\boldsymbol{x} \in X_T^\epsilon) = \bar{\beta}_t. \tag{6}$$

Notably, this process is independent of the random point set $X_0$, i.e., $\forall t : q(X_t^\epsilon|X_0) = q(X_t^\epsilon)$.

We present a visual depiction of the two forward processes in Figure 2. Finally, given that $\forall t : X_t = X_t^{\text{thin}} \bigcup X_t^\epsilon$ it follows that for $\lim_{t \to T} \bar{\alpha}_t = 0$ and $\lim_{t \to T} \bar{\beta}_t = 1$ the *stationary distribution* is $q(X_T|X_0) = p_{\text{noise}}(X)$, which can be seen by applying the superposition property and finding the intensity of $X_t|X_0$ to be $\bar{\alpha}_t \lambda_0 + \bar{\beta}_t \lambda^\epsilon$. To summarize, the forward process gradually removes points from the original point set $X_0 \sim p_{\text{data}}(X)$ while progressively adding points of a noise point set $X_T \sim p_{\text{noise}}(X)$, stochastically interpolating between data and noise.

### 3.2 REVERSE PROCESS

To generate samples from our diffusion model, i.e., $X_T \to \cdots \to X_0$, we need to learn how to reverse the forward process by approximating the posterior $q(X_t|X_0, X_{t+1})$ with a model $p_\theta(X_t|X_{t+1})$. We will start by deriving the posterior $q(X_t|X_0, X_{t+1})$ from the forward process $q(X_{t+1}|X_t)$ and then show how to parameterize and train $p_\theta(X_t|X_{t+1})$ to approximate the posterior.

Since the forward process consists of two independent processes (*thinning* and *noise*) and noticing that $X_{t+1}^{\text{thin}} = X_0 \bigcap X_{t+1}$ and $X_{t+1}^\epsilon = X_{t+1} \setminus X_0$, the posterior can be derived in two parts:

**Thinning posterior:** Since all points in $X_{t+1}^{\text{thin}}$ have been retained from $t = 0$, it follows that $X_{t+1}^{\text{thin}} \subseteq X_t^{\text{thin}}$. Then for each point in $\boldsymbol{x} \in X_0 \setminus X_{t+1}^{\text{thin}}$, we derive then posterior using Bayes' theorem, applying Equation 3, Equation 4 and the Markov property:

$$q(\boldsymbol{x} \in X_t^{\text{thin}}|\boldsymbol{x} \notin X_{t+1}^{\text{thin}}, \boldsymbol{x} \in X_0) = \frac{q(\boldsymbol{x} \notin X_{t+1}^{\text{thin}}|\boldsymbol{x} \in X_t^{\text{thin}})q(\boldsymbol{x} \in X_t^{\text{thin}}|\boldsymbol{x} \in X_0)}{q(x \notin X_{t+1}^{\text{thin}}|\boldsymbol{x} \in X_0)} \tag{7}$$

$$= \frac{(1 - \alpha_{t+1})\bar{\alpha}_t}{(1 - \bar{\alpha}_{t+1})} = \frac{\bar{\alpha}_t - \bar{\alpha}_{t+1}}{1 - \bar{\alpha}_{t+1}}. \tag{8}$$

Thus, we can sample $X_t^{\text{thin}}$ by superposition of $X_{t+1}^{\text{thin}}$ and thinning $X_0 \setminus X_{t+1}^{\text{thin}}$ with Equation 7.

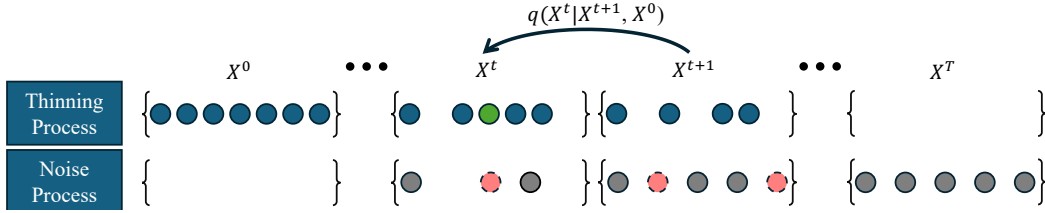

Figure 3: The posterior reverses the stochastic interpolation of $X_0 \rightarrow X_T$ of the forward process by adding back thinned points from the thinning process and thinning point added in the noise process.

**Noise posterior:** Following the reverse thinning interpretation of the noise process, each point in $X_t^\epsilon$ must have been in both $X_{t+1}^\epsilon$ and $X_T^\epsilon$. Hence, we derive the posterior for each point in $X_{t+1}^\epsilon$ to still be in $X_t^\epsilon$ by following Equation 6, along with the fact that $X_t^\epsilon$ is independent from $X_0$:

$$q(\boldsymbol{x} \in X_t^\epsilon | \boldsymbol{x} \in X_{t+1}^\epsilon, \boldsymbol{x} \notin X_0) = \frac{q(\boldsymbol{x} \in X_{t+1}^\epsilon | \boldsymbol{x} \in X_t^\epsilon) q(\boldsymbol{x} \in X_t^\epsilon)}{q(\boldsymbol{x} \in X_{t+1}^\epsilon)} \tag{9}$$

$$= \frac{1 \cdot \bar{\beta}_t}{(\bar{\beta}_{t+1})} = \frac{\bar{\beta}_t}{\bar{\beta}_{t+1}}. \tag{10}$$

Thus, we can sample $X_t^\epsilon$ by thinning $X_{t+1}^\epsilon$ with probability $(1 - \frac{\bar{\beta}_t}{\bar{\beta}_{t+1}})$.

**Parametrization:** Given $X_0$ and $X_T$, the derived posterior can reverse the noising process to generate $X_0$. However, to generate a new approximate sample $X_0 \sim p_{\text{data}}(X)$, we need to be able to sample from the posterior $q(X_t | X_0, X_{t+1})$ without knowing $X_0$. For this reason we approximate the posterior with a model $p_\theta(X_t | X_{t+1})$, where we choose $p_\theta(X_t | X_{t+1}) = \int q(X_t | \widetilde{X}_0, X_{t+1}) p_\theta(\widetilde{X}_0 | X_{t+1}) \, d\widetilde{X}_0$ and training a neural network $p_\theta(\widetilde{X}_0 | X_t)$ to approximate $X_0 | X_{t+1}$ for each $t + 1$.

To effectively train this model, we have to condition our model $p_\theta(\widetilde{X}_0 | X_t)$ on $X_t$. We propose to embed the points $\boldsymbol{x} \in X_t$ permutation invariant with a Transformer encoder with full attention and apply a sinusoidal embedding to embed $n = |X_t|$ and $t$. Then, to probilistically predict $X_0 | X_t$, we make use of the following case distinction for $X_t^{\text{thin}} = X_0 \bigcap X_t$ and $X_0 \setminus X_t$:

*First*, predicting the retained points in $X_t$, i.e., the intersection of $X_0$ and $X_t$, is a binary classification task for which we train a multi-layer-perceptron (MLP) $g_\theta(\boldsymbol{x} \in X_t^{\text{thin}} | X_t, t)$ with binary cross entropy loss $\mathcal{L}_{\text{BCE}}$. *Second*, the thinned points in $X_0$, i.e., $X_0 \setminus X_t$, is a point set $N$, which can be represented by its counting measure, as a mixture of $n$ Dirac measures:

$$N = \sum_{i=1}^{n} \delta_{X_i}. \tag{11}$$

In A.2, we prove that any finite mixture of Dirac deltas, such as $N$, can be approximated by an $L^2$ function in $L^2(D, \mu)$ for any metric space $D$. In Euclidean spaces, we approximate the Dirac measure with a mixture of multivariate Gaussian distributions with diagonal covariance matrices. Note that the multivariate Gaussian density function is a standard approximation of the Dirac delta function and, as the determinant of a diagonal covariance matrix $\boldsymbol{\Sigma} := \sigma \boldsymbol{I}$ approaches zero, the Gaussian increasingly resembles the Dirac delta (See Equation A.2). We parameterize the number of points to sample $n_\theta$ and the components of the mixture — weights $\boldsymbol{w}_\theta$, mean $\boldsymbol{\mu}_\theta$ and diagonal covariance matrix $\boldsymbol{\Sigma}_\theta$ — with an MLP $f_\theta$ and train it with the negative log likelihood $\mathcal{L}_{\text{NLL}}$.

Lastly, to ensure the expected number of points at any time $t$ throughout the diffusion process is constant, we use $\bar{\alpha}_t = 1 - \bar{\beta}_t$ and a noise process with a constant intensity such that $\int_A \lambda^\epsilon = E[N(A)]$ for the bounded Borel set $A$ that represents our domain.

### 3.3 SAMPLING PROCEDURE

**Unconditional sampling:** Starting from a sample $X_T$ of the noise distribution, we apply our POINT SET DIFFUSION model to sample a new $X_0$ over $T$ steps. We start by sampling $X_T \sim \lambda_\epsilon$ and

then for all $t \in (T, \dots, 1)$ sample $\widetilde{X}_0 \sim p_\theta(X_0|X_t)$ to subsequently apply the denoising posterior $q(X_{t-1}|\widetilde{X}_0, X_t)$ and attain $X_{t-1}$. Finally, at step 1 we sample $\widetilde{X}_0 \sim p_\theta(X_0|X_1)$. We present the extended sampling algorithm in Algorithm 2.

**Conditional sampling:** Let $C : D \rightarrow \{0, 1\}$ be a conditioning mask on our metric space $D$, where we define the masking of a subset $X \subseteq D$ as $C(X) \coloneqq \{\boldsymbol{x} \in X | C(\boldsymbol{x}) = 1\}$ and its complement as $C'(X) \coloneqq \{\boldsymbol{x} \in X | C(\boldsymbol{x}) = 0\}$. Then, we can leverage our POINT SET DIFFUSION model to conditionally generate random point sets outside the conditioning mask by applying Algorithm 1:

---

**Algorithm 1** Conditional sampling

---

**Require:** $X_0^c = C(X_0)$
1: $X_T \sim \lambda_\epsilon$
2: **for** $t = T, \dots, 1$ **do**
3: $\quad \widetilde{X}_0 \sim p_\theta(X_0|X_t)$
4: $\quad \widetilde{X}_{t-1} \sim q(X_{t-1}|\widetilde{X}_0, X_t)$ $\quad$ (**reverse 3.2**)
5: $\quad X_{t-1}^c \sim q(X_{t-1}^c|X_0^c)$ $\quad\quad$ (**forward 3.1**)
6: $\quad X_{t-1} = C'(\widetilde{X}_{t-1}) \cup C(X_{t-1}^c)$
7: **end for**
8: **return** $C'(X_0)$

---

Figure 4: Examples of conditioning masks for $\mathbb{R}_{\geq 0}$ and $\mathbb{R}^2$.

Thus, following this sampling procedure, we can generate conditional samples for any conditioning mask $C$, where we represent some illustrative conditioning masks for bounded sets on $\mathbb{R}_{\geq 0}$ and $\mathbb{R}^2$ depicting temporal forecasting, history prediction and general imputation tasks in Figure 4.

## 4 RELATED WORK

Since large parts of the real-world can be effectively captured by Euclidean spaces, point processes have mainly been defined on spatial and temporal dimensions, represented by an Euclidean space. Hence, for this discussion of the related work, we will focus on unordered and ordered point processes on Euclidean spaces, mainly SPPs, TPPs and STPPs. For completeness, we want to mention traditional parametric point processes defined on manifolds, such as determential point processes (Berman, 2008; Katori & Shirai, 2022) and cluster point processes (Bogachev & Daletskii, 2013).

**Unordered Point Processes (SPP):** Modeling a permutation-invariant intensity for unordered point sets that captures complex interactions while remaining efficient for sampling is challenging (Daley & Vere-Jones, 2007), seemingly limiting the development of machine-learning-based models for SPPs. Classical models like the Poisson Point Process (Kingman, 1992) use either homogeneous or inhomogeneous intensity functions across space. More flexible models, such as Cox processes (Cox, 1955), and specifically the popular *Log-Gaussian Cox Process* (Jesper Møller, 1998), extend this by modeling the intensity function through a doubly stochastic process, allowing for flexible spatial inhomogeneity. A recent approach, the *Regularized Method* by Osama et al. (2019), parameterizes a spatial Poisson process on a hexagonal grid with splines, offering out-of-sample guarantees. However, these methods often rely on spatial discretization and simple parametric forms and some require separate intensity estimates for each point set, limiting their ability to capture the underlying distribution across different samples (Daley & Vere-Jones, 2007; Osama et al., 2019).

**Ordered point processes (TPP and STPP):** The causal ordering of time enables the parametrization of a conditional intensity, which classically is being modeled with parametric functions, where the Hawkes Process (Hawkes, 1971) is the most widely used model and captures point interaction patterns like self-excitation. Given the sequential nature of ordered point process, a variety of Machine Learning based approaches for TPPs and STPPs have been proposed (see Shchur et al. (2021) for a review on neural TPPs). Where recurrent neural network- (Du et al., 2016; Shchur et al., 2020a) and transformer-based encoders (Zhang et al., 2020a; Zuo et al., 2020; Chen et al., 2020) are leveraged to encode the history and neurally parameterized Hawkes (Zhou & Yu, 2023; Zhang

et al., 2020a; Zuo et al., 2020), parametric density functions (Du et al., 2016; Shchur et al., 2020a), mixtures of kernels (Okawa et al., 2019; Soen et al., 2021; Zhang et al., 2020b; Zhou et al., 2022), neural networks (Omi et al., 2019; Zhou & Yu, 2023), Gaussian diffusion (Lin et al., 2022; Yuan et al., 2023) and normalizing flows (Chen et al., 2020; Shchur et al., 2020b) have been proposed to (non)-parametrically decode the conditional density or intensity of the next event.

**Differences to ADD-THIN (Lüdke et al., 2023):** Since our method is closely related to ADD-THIN, we want to highlight their key methodological differences. While ADD-THIN proposed to leverage the thinning and superposition properties to define a diffusion process for TPPs, POINT SET DIFFUSION generalizes this idea to define a diffusion-based latent variable model for point processes on general metric spaces. In doing so, we disentangle the superposition and thinning to attain two independent processes to allow for more explicit control and define the diffusion model independent of the intensity function as a stochastic interpolation of point sets. Furthermore, ADD-THIN has to be trained for specific conditioning tasks, while we show how to condition our unconditional POINT SET DIFFUSION model for arbitrary conditioning tasks on the metric space. Lastly, POINT SET DIFFUSION and its parametrization are agnostic to the ordering of points, making it applicable to model the general class of point processes on any metric space, including, for example, SPPs.

## 5 EXPERIMENTS

Although point processes are fundamentally generative models, the standard evaluation method relies on reporting the negative log-likelihood (NLL) on a hold-out test set, effectively reducing the evaluation to single-event predictions for STPPs and TPPS. However, this approach presents two key issues. *First*, computing the NLL depends on the specific implementation and parameterization of the (conditional) intensity function and is intractable for many models, necessitating approximations using Monte Carlo methods, numerical integration, or the evidence lower bound (ELBO). This complicates fair comparisons between models. *Second*, evaluating the likelihood conditioned on ground-truth history, does not necessarily reflect how well a model captures the data distribution or its ability to perform on complex conditional generation tasks (Shchur et al., 2021). To overcome these limitations, we evaluate the generative capabilities of our proposed POINT SET DIFFUSION model by benchmarking it on a range of unconditional and conditional generation tasks for SPP and STPP. Further, we compare our model's performance with the state-of-the-art TPP model ADD-THIN in A.6. Details of our model's training and the hyperparameters are in A.4, while all baselines are trained reproducing their reported NLL using their proposed hyperparameters and code.

### 5.1 DATA

We follow Chen et al. (2021) and evaluate our model on four benchmark datasets with their proposed pre-processing and splits: three real-world datasets — *Japan Earthquakes* (U.S. Geological Survey, 2024), *New Jersey COVID-19* Cases (The New York Times, 2024), and *Citibike Pickups* (Citi Bike, 2024) —and one synthetic dataset, *Pinwheel*, based on a multivariate Hawkes process (Soni, 2019).

### 5.2 METRICS

To evaluate both unconditional and conditional tasks, we compute distances between point process distributions and individual point sets, assuming the space is normed, and all points are bounded, i.e., $\forall i, \boldsymbol{x}_i \in [-1, 1]^d$. We use the following metrics in our evaluation:

**Sequence Length (SL):** To compare the length distribution of point sets, we report the Wasserstein distance between the two categorical distributions. For conditional tasks, we compare the length of the generated point set to the ground truth by reporting the Mean Absolute Error (MAE).

**Counting Distance (CD):** Xiao et al. (2017) introduced a Wasserstein distance for ordered TPPs based on Birkhoff's theorem. We generalize this counting distance to higher-dimensional ordered Euclidean spaces (e.g., STPPs) using the $L_1$ distance:

$$CD(X, Y) = \frac{1}{d} \sum_{i=1}^{k} ||\boldsymbol{x}_i - \boldsymbol{y}_i||_1 + \sum_{j=k+1}^{l} ||U - \boldsymbol{y}_j||_1, \qquad (12)$$

Table 1: Density estimation results on the hold-out test set for SPPs, averaged over three random seeds (**bold** best and underline second best).

| | Earthquakes | | Covid NJ | | Citybike | | Pinwheel | |
|---|---|---|---|---|---|---|---|---|
| | SL($\downarrow$) | MMD($\downarrow$) | SL($\downarrow$) | MMD($\downarrow$) | SL($\downarrow$) | MMD($\downarrow$) | SL($\downarrow$) | MMD($\downarrow$) |
| LOG-GAUSSIAN COX | 0.047 | 0.214 | 0.209 | 0.340 | 0.104 | 0.336 | **0.017** | 0.285 |
| REGULARIZED METHOD | 2.361 | 0.391 | 0.255 | 0.411 | 0.097 | 0.342 | 0.039 | 0.411 |
| POINT SET DIFFUSION | **0.038** | **0.173** | **0.199** | **0.268** | **0.056** | **0.092** | **0.017** | **0.099** |

Table 2: Conditional generation results on the hold-out test set for SPP, averaged over three random seeds (**bold** best).

| | Earthquakes | | Covid NJ | | Citybike | | Pinwheel | |
|---|---|---|---|---|---|---|---|---|
| | MAE($\downarrow$) | WD($\downarrow$) | MAE($\downarrow$) | WD($\downarrow$) | MAE($\downarrow$) | WD($\downarrow$) | MAE($\downarrow$) | WD($\downarrow$) |
| REGULARIZED METHOD | 30.419 | 0.162 | 16.075 | 0.148 | 7.740 | 0.115 | 3.547 | 0.150 |
| POINT SET DIFFUSION | **4.651** | **0.106** | **5.056** | **0.119** | **3.498** | **0.085** | **2.256** | **0.122** |

where $X = \{\boldsymbol{x}_i\}_{i=1}^{k}$ and $Y = \{\boldsymbol{y}_i\}_{i=1}^{l}$ are two ordered samples from a point process on a metric space of dimensionality $d$, i.e. $D \subseteq \mathbb{R}^d$. Further, $U \coloneqq (\boldsymbol{u}_1, \ldots, \boldsymbol{u}_d)$ represents the upper bounds of the metric space $D$ along each dimension and we assume, without loss of generality, $l \geq k$.

**Wasserstein Distance (WD):** An instance of a Point Process is itself a stochastic process of points in space. Hence, we can compute a distance between two point sets based on the Wasserstein distance on the metric space $D \subseteq \mathbb{R}^d$ between the two sets of points.

**Maximum Mean Discrepancy (MMD)** (Gretton et al., 2012)**:** The kernel-based statistic test compares two distributions based on a distance metric; we use the WD for SPPs and CD for STPP.

## 5.3 SPATIAL POINT PROCESSES

We evaluate our model's ability to capture the distribution of spatial point processes (SPP) by benchmarking it against two methods. The first is the widely used LOG-GAUSSIAN COX PROCESS (Jesper Møller, 1998), a doubly stochastic model that parameterizes the intensity function using a Gaussian process. The second is the REGULARIZED METHOD (Osama et al., 2019), leveraging a regularized criterion to infer predictive intensity intervals, offering out-of-sample prediction guarantees and enabling conditional generation.

**Unconditional Generation (Density Estimation):** In this experiment, we generate 1,000 unconditional samples from each model and compare their distribution to a hold-out test set using the WD-SL and WD-MMD metrics. As shown in Table 1, our POINT SET DIFFUSION model consistently generates samples most closely matching the data distribution across all datasets. While the baseline models perform reasonably well in capturing the count distributions for most datasets, their reliance on spatial discretization and smoothness properties of the intensity function limit their ability to capture the complex spatial patterns in the data, as reflected by higher WD-MMD scores.

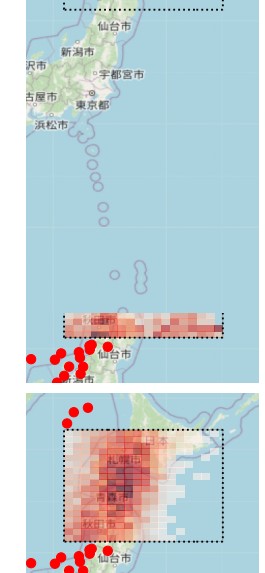

Figure 5: SPP conditioning task: top ground truth, middle REGULARIZED METHOD and bottom POINT SET DIFFUSION.

**Conditional Generation:** To assess POINT SET DIFFUSION's ability to solve spatial conditioning tasks, we sample 50 random bounding boxes (with widths uniformly sampled between 1/8 and 3/8 of the metric space) for imputation on the hold-out test set, and report the results in Table 2. The REGULARIZED METHOD fits a spatial Poisson model with out-of-sample accuracy guarantees and has been shown by Osama et al. (2019) to outperform the LOG-GAUSSIAN COX PROCESS on interpolation and extrapolation tasks. However, we find that the REGULARIZED METHOD's reliance on predicting a smooth and discretized intensity function conditioned on neighboring areas leads to inaccurate imputations when the adjacent regions contain significantly different numbers of points (see the hexagonal discretization structure

Table 3: Density estimation results on the hold-out test set for STPP, averaged over three random seeds (**bold** best and underline second best).

| | Earthquakes | | Covid NJ | | Citybike | | Pinwheel | |
|---|---|---|---|---|---|---|---|---|
| | SL($\downarrow$) | MMD($\downarrow$) | SL($\downarrow$) | MMD($\downarrow$) | SL($\downarrow$) | MMD($\downarrow$) | SL($\downarrow$) | MMD($\downarrow$) |
| DEEPSTPP | 0.105 | 0.266 | **0.169** | 0.166 | 3.257 | 0.677 | 1.067 | 0.197 |
| DIFFSTPP | 0.088 | 0.064 | 0.332 | 0.146 | 0.560 | 0.611 | 0.196 | 0.055 |
| AUTOSTPP | 0.073 | 0.062 | 0.364 | 0.280 | 0.598 | 0.331 | 0.127 | 0.147 |
| POINT SET DIFFUSION | **0.042** | **0.023** | 0.189 | **0.043** | **0.032** | **0.020** | **0.023** | **0.020** |

Table 4: Forecasting results on the hold-out test set for STPP, averaged over three random seeds (**bold** best and underline second best).

| | Earthquakes | | Covid NJ | | Citybike | | Pinwheel | |
|---|---|---|---|---|---|---|---|---|
| | MAE($\downarrow$) | CD($\downarrow$) | MAE($\downarrow$) | CD($\downarrow$) | MAE($\downarrow$) | CD($\downarrow$) | MAE($\downarrow$) | CD($\downarrow$) |
| DEEPSTPP | 10.154 | 11.211 | **6.264** | **8.492** | 127.968 | 125.747 | 18.651 | 15.792 |
| DIFFSTPP | 16.027 | 17.466 | 18.822 | 14.302 | 7.516 | 8.460 | 14.461 | 13.062 |
| POINT SET DIFFUSION | **7.407** | **10.458** | 7.293 | 10.865 | **5.928** | **7.225** | **6.341** | **6.437** |

and smoothness in Figure 5). This issue is exacerbated by not capturing a shared intensity function across point sets, making it difficult for the REGULARIZED METHOD to handle non-smooth spatial patterns, such as varying inhomogeneous intensities shared across multiple point sets. This highlights a core limitation of SPP models that rely on instance-specific intensity functions.

## 5.4 SPATIO-TEMPORAL POINT PROCESSES

For STPPs, we compare our model to three state-of-the-art STPP models, which parameterize an autoregressive intensity function, to assess the model's ability to capture the point process distribution. DEEPSTPP (Zhou et al., 2022) uses a latent variable framework to non-parametrically model the conditional intensity based on kernels. DIFFSTPP (Yuan et al., 2023) is based on a diffusion model approximating the conditional intensity. Lastly, AUTOSTPP (Zhou & Yu, 2023) uses the automatic integration for neural point processes, presented by Lindell et al. (2021), to parameterize a generalized spatiotemporal Hawkes model.

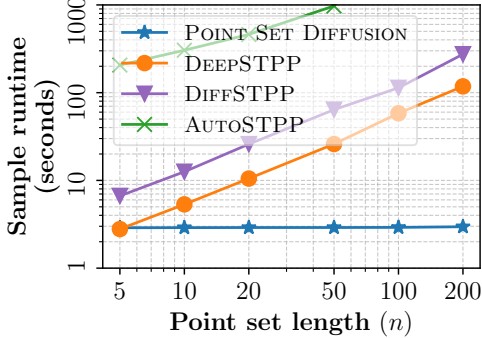

Figure 6: STPP runtime for sampling $n$ points.

**Sampling Runtime:** We report the median sampling runtime over ten runs generating ten point sets of length $n$ on an NVIDIA A100-PCIE-40GB for all STPP models in Figure 6. POINT SET DIFFUSION maintains a nearly constant runtime for all point set lengths as it generates all points in parallel, whereas autoregressive baselines, due to their sequential sampling, exhibit a linear relationship between runtime and $n$.

**Unconditional Generation (Density Estimation):** We evaluate the performance of each model by comparing the WD-SL and CS-MMD between the hold-out test set and 1,000 samples generated by the trained models, as shown in Table 3. Again, the POINT SET DIFFUSION model best captures the distribution of the point process distribution for all datasets. The autoregressive intensity functions of the baseline models fail to generate point sets that align closely with the data distribution for most datasets, as reflected in the differences in the WD-SL and CD-MMD metrics compared to POINT SET DIFFUSION. While these baselines are trained to predict the next event given a history window, they struggle to unconditionally sample realistic point sets when starting from an empty sequence. Consequently, this highlights our argument that the standard evaluation based on NLL is insufficient to assess the true generative capacity of point process models.

**Conditional Generation (Forecasting):** Forecasting future events based on historical data is a challenging and a fundamental task for STPP models. To evaluate this capability, we uniformly sampled 50 random starting times from the interval $[\frac{5}{8}U_{time}, \frac{7}{8}U_{time}]$, where $U_{time}$ is the maximum time, for

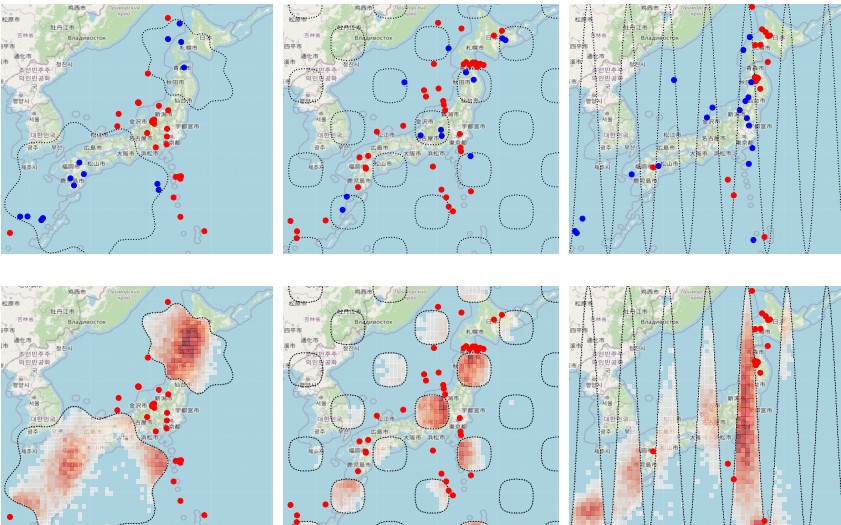

Figure 7: Complex spatial conditioning tasks solved with POINT SET DIFFUSION: Top condition and ground truth data, bottom density plots for predictions.

each point set in the hold-out test set. The results are detailed in Table 4.[2] The autoregressive baselines, trained to predict the next event based on history, achieve good forecasting results for most datasets, one even surpassing POINT SET DIFFUSION on the Covid NJ dataset. Still, our unconditional model outperforms the autoregressive baselines across all other datasets.

## 5.5 OTHER CONDITIONING TASK

Since the STPP baselines are autoregressive models, they are limited to forecasting tasks. However, our model can generate conditional samples for any conditioning mask $C$ on our metric space. To showcase this feature, we present a few visual examples of complex conditioning tasks in Figure 7.

## 6 CONCLUSION

To model general point processes on metric spaces, we present POINT SET DIFFUSION, a novel diffusion-based latent variable model. We derive POINT SET DIFFUSION as a stochastic interpolation between data point sets and noise point sets governed by the thinning and superposition properties of random point sets. Thereby, we attain a very flexible, unconditional Point Process model that can be conditioned for arbitrary condition masks on the metric space and allows for efficient and parallel sampling of entire point sets without relying on the (conditional) intensity function. In conditional and unconditional experiments on synthetic and real-world SPP, TPP and STPP data, we demonstrate that POINT SET DIFFUSION achieves state-of-the-art performance while allowing for up to orders of magnitude faster sampling.

We have introduced a generative model for point processes on general metric spaces, prioritizing generality, scalability, and flexibility to address key limitations of intensity-based models. While this enables unconditional modeling and flexible generation for arbitrary conditional tasks on any metric space, it does not permit interpreting the conditional intensity or its parameters. Thus, for inference applications of STPPs or TPPs that require estimating the conditional intensity of the next event, point process models that directly approximate this conditional intensity are better suited. Ultimately, with POINT SET DIFFUSION, we have presented a novel set modeling approach and would be interested to see how future work explores its limitations on other (high-dimensional) metric (e.g., Riemannian manifolds), topological and discrete spaces with potential applications extending beyond traditional point sets including but not limited to natural language and graphs.

---

[2]AutoSTPP is not included in this analysis due to its prohibitively slow sampling speed (see Figure 6 and the limitations discussed in Zhou & Yu (2023)), which made it impractical to sample the 50 forecast windows for all instances in the test set within a reasonable timeframe.

## ACKNOWLEDGEMENT

This research was supported by the German Research Foundation, grant GU 1409/3-1.

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

## A  APPENDIX

### A.1  POINT PROCESS PROPERTIES

The thinning and superposition properties have been proved by other works for different versions of point processes. For completeness and generality, we prove them for a general Borel set $A$. To apply these proofs for SPPs consider $A \subseteq \mathcal{S}$, where $\mathcal{S}$ is a metric space in $\mathbb{R}^d$ and for STPPs consider $A \subseteq [0, T] \times \mathcal{S}$, where $T > 0$.

**Superposition:**  **Proof.** It is straightforward to obtain the superposition expectation measure from Equation 1:

$$\mu(A) = \mathbb{E}[N(A)] = \mathbb{E}[N_1(A) + N_2(A)] = \mathbb{E}[N_1(A)] + \mathbb{E}[N_2(A)] = \mu_1(A) + \mu_2(A). \quad (13)$$

Then, every point process has an intensity of $\lambda_1$ and $\lambda_2$ for each of the expectation measures $\mu_1$ and $\mu_2$, respectively. Therefore, taking the right-hand side of Equation 1, we obtain the following intensity function for the superposition of point processes:

$$\mu(A) = \mu_1(A) + \mu_2(A) = \int_A \lambda_1(x)dx + \int_A \lambda_2(x)dx = \int_A \lambda_1(x) + \lambda_2(x)dx. \quad (14)$$

This states that the density function for expectation measure $\mu$ is $\lambda := \lambda_1 + \lambda_2$, and concludes the proof for the superposition property of intensities for point processes. $\qquad \square$

**Thinning:**  **Proof.** For this property, we need to assume that the singletons are simple, so we can only have one point at each position: $N(\{x\}) \leq 1$; these point processes are called *simple*. Simple point processes can be represented as a sum of Dirac measures at the random points $X_i \in \mathcal{S}$:

$$N = \sum_i \delta_{X_i}. \quad (15)$$

The previous assumption on singletons makes the sum above a finite sum. If $Z_i \in \{0, 1\}$ are Bernoulli random variables with a success probability $p$ we can define a random thinning process as the superposition of the following point processes:

$$N_1 = \sum_i Z_i \delta_{X_i}. \quad (16)$$

$$N_2 = \sum_i (1 - Z_i) \delta_{X_i}. \quad (17)$$

Since there are only two options for the Bernoulli random variable, it holds that the superposition of the point processes defined in Equations 16 and 17 are equivalent to the original point process, i.e., $N = N_1 + N_2$.

Given that all $Z_i \sim Bern(p)$ are i.i.d., we obtain a conditional probability distribution on the thinned point process $N_1(A)|N(A) = n \sim Binom(n, p)$. And by the law of total expectation, we derive:
$$\mu_1(A) = \mathbb{E}[N_1(A)] = \mathbb{E}\big[\mathbb{E}[N_1(A)|N(A)]\big] = \mathbb{E}[N(A)p] = \mu(A)p. \quad (18)$$
We can write the terms of the equation before in terms of the intensity measure of the point process:

$$\mu_1(A) = p \cdot \mu(A) = p \int_A \lambda(x)dx = \int_A p\lambda(x)dx. \quad (19)$$

Hence, the equation above implies that the intensity of the new point process $N_1$, which keeps the points of the original point process $N$ with probability $p$, is $p\lambda$.

By the property of superposition, since $N = N_1 + N_2$, then $\lambda = p\lambda + (1 - p)\lambda$. Therefore, the intensity of the point process $N_2$, containing the thinned points, is $(1 - p)\lambda$.

This proves that, in the opposite case, when removing points with probability $p$ from a given point process with intensity $\lambda$, the intensity of the point process with the points kept after thinning is $(1 - p)\lambda$. $\qquad \square$

## A.2 Approximation of Mixture of Dirac Delta Functions by $L^2$-Functions

**Definition 1 (Dirac delta function)** *Let $(D, d, \mu)$ be a general metric space equipped with a measure $\mu$. A Dirac delta function $\delta_{\boldsymbol{x}}$ at a point $\boldsymbol{x} \in D$ is defined as a distribution such that for any test function $f$:*

$$\int_D f(\boldsymbol{y})\delta_{\boldsymbol{x}}(\boldsymbol{y})d\mu(\boldsymbol{y}) = f(\boldsymbol{x}). \tag{20}$$

**Theorem 1** *Let $f_M(\boldsymbol{y})$ be a finite mixture of Dirac deltas:*

$$f_M(\boldsymbol{y}) = \sum_{i=1}^n w_i \delta_{\boldsymbol{x}_i}(\boldsymbol{y}), \tag{21}$$

*where $\boldsymbol{x}_1, \ldots, \boldsymbol{x}_n \in D$ are points in the metric space, and $w_i \in \mathbb{R}$ are weights associated with each Dirac delta function. Then, this finite mixture of Dirac deltas $f_M$ can be approximated by $L^2$ functions in $L^2(D, \mu)$.*

**Proof.** We use a sequence of smooth functions that approximate each Dirac delta in the mixture and then show that this approximation converges in the $L^2$-norm.

Firstly, we show how to approximate Dirac delta functions. Let us consider a family of smooth functions $\phi_\epsilon(\boldsymbol{x})$ (such as bump functions or mollifiers) that approximate the Dirac delta function $\delta_{\boldsymbol{x}}$ as $\epsilon \to 0$. These functions $\phi_\epsilon(\boldsymbol{x} - \boldsymbol{x}_i)$ are supported near $\boldsymbol{x}_i$ and satisfy:

$$\lim_{\epsilon \to 0} \phi_\epsilon(\boldsymbol{x} - \boldsymbol{x}_i) = \delta_{\boldsymbol{x}_i}(\boldsymbol{x}). \tag{22}$$

In particular, for any test function $f$, we have:

$$\int_D f(\boldsymbol{y})\phi_\epsilon(\boldsymbol{y} - \boldsymbol{x}_i)d\mu(\boldsymbol{y}) \to f(\boldsymbol{x}_i) \quad \text{as} \quad \epsilon \to 0. \tag{23}$$

Hence, $\phi_\epsilon(\boldsymbol{x} - \boldsymbol{x}_i)$ has a similar property as the one of Dirac deltas given in Equation 20 and serves as an approximation of the Dirac delta $\delta_{\boldsymbol{x}_i}(\boldsymbol{x})$ for a small $\epsilon$.

Secondly, we approximate the mixture of Dirac deltas $f_M$ by a function in $L^2(D, \mu)$ using the same $\phi_\epsilon(\boldsymbol{x})$-based approximation for each Dirac delta, defining:

$$f_\epsilon(\boldsymbol{y}) = \sum_{i=1}^n w_i \phi_\epsilon(\boldsymbol{y} - \boldsymbol{x}_i). \tag{24}$$

Each term $\phi_\epsilon(\boldsymbol{y} - \boldsymbol{x}_i)$ is a smooth approximation of the corresponding Dirac delta $\delta_{\boldsymbol{x}_i}(\boldsymbol{y})$, and the sum represents the approximation of the entire mixture of Diracs.

Thirdly, we show that the sequence $f_\epsilon$ converges to $f_M$ in the $L^2$-norm, i.e., that:

$$\lim_{\epsilon \to 0} \|f_\epsilon - f_M\|_{L^2(D,\mu)} = 0. \tag{25}$$

Since $f_M$ is a sum of Dirac deltas, it is not directly in $L^2(D, \mu)$, but its approximation $f_\epsilon$ is because each $\phi_\epsilon$ is a smooth function and smooth functions with compact support are in $L^2(D, \mu)$.

We compute now the squared $L^2$ norm of the difference in Equation 25:

$$\|f_\epsilon - f_M\|_{L^2(D,\mu)}^2 = \int_D |f_\epsilon(\boldsymbol{y}) - f_M(\boldsymbol{y})|^2 \, d\mu(\boldsymbol{y}). \tag{26}$$

Note that the squared difference of $f_\epsilon$ and $f_M$ in the above equation will have quadratic and crossed terms. However, we can neglect the crossed terms: $2\sum_{i<j} w_i w_j \int_D \left(\phi_\epsilon(\boldsymbol{y} - \boldsymbol{x}_i) - \delta_{\boldsymbol{x}_i}(\boldsymbol{y})\right)\left(\phi_\epsilon(\boldsymbol{y} - \boldsymbol{x}_j) - \delta_{\boldsymbol{x}_j}(\boldsymbol{y})\right) d\mu(\boldsymbol{y})$, since every smooth function $\phi_\epsilon(\boldsymbol{y} - \boldsymbol{x}_i)$ is concentrated near $\boldsymbol{x}_i$ and terms involving different indices do not contribute to the limit.

Hence, we can simplify the norm in Equation 26 into the sum of the individual terms:

$$\|f_\epsilon - f_M\|_{L^2(D,\mu)}^2 = \sum_{i=1}^n \int_D w_i^2 |(\phi_\epsilon(\boldsymbol{y} - \boldsymbol{x}_i) - \delta_{\boldsymbol{x}_i}(\boldsymbol{y}))|^2 d\mu(\boldsymbol{y}). \tag{27}$$

For every $i$, the term $\int_D |\phi_\epsilon(\boldsymbol{y} - \boldsymbol{x}_i) - \delta_{\boldsymbol{x}_i}(\boldsymbol{y})|^2 d\mu(\boldsymbol{y})$ becomes small as $\epsilon \to 0$, because by construction $\phi_\epsilon(\boldsymbol{y} - \boldsymbol{x}_i) \to \delta_{\boldsymbol{x}_i}(\boldsymbol{y})$ in the sense of distributions. Thus, by the properties of $\phi_\epsilon$, we conclude that:

$$\lim_{\epsilon \to 0} \|f_\epsilon - f_M\|_{L^2(D,\mu)} = 0. \tag{28}$$

$\square$

**Lemma 1** *Let $p(\boldsymbol{x}; \boldsymbol{\mu}, \boldsymbol{\Sigma})$ be the probability density function (PDF) of a multivariate Gaussian distribution. Then $p \in L^2(\mathbb{R}^d)$.*

**Proof.** The PDF of a multivariate Gaussian distribution in $\mathbb{R}^d$ with mean vector $\boldsymbol{\mu} \in \mathbb{R}^d$ and covariance matrix $\boldsymbol{\Sigma}$ (which is positive definite) is given by:

$$p(\boldsymbol{x}; \boldsymbol{\mu}, \boldsymbol{\Sigma}) = \frac{1}{(2\pi)^{n/2}|\boldsymbol{\Sigma}|^{1/2}} \exp\left(-\frac{1}{2}(\boldsymbol{x} - \boldsymbol{\mu})^T \boldsymbol{\Sigma}^{-1}(\boldsymbol{x} - \boldsymbol{\mu})\right), \tag{29}$$

where $\boldsymbol{x} \in \mathbb{R}^d$, $|\boldsymbol{\Sigma}|$ is the determinant of the covariance matrix $\boldsymbol{\Sigma}$, and $\boldsymbol{\Sigma}^{-1}$ is the inverse of the covariance matrix. We show that $\|p\|_{L^2} = \left(\int_{\mathbb{R}^d} |p(\boldsymbol{x})|^2 \, d\boldsymbol{x}\right)^{1/2}$ is finite.

We need to compute the following integral:

$$\int_{\mathbb{R}^d} p(\boldsymbol{x})^2 \, d\boldsymbol{x} = \frac{1}{(2\pi)^n |\boldsymbol{\Sigma}|} \int_{\mathbb{R}^d} \exp\left(-(\boldsymbol{x} - \boldsymbol{\mu})^T \boldsymbol{\Sigma}^{-1}(\boldsymbol{x} - \boldsymbol{\mu})\right) d\boldsymbol{x}. \tag{30}$$

To simplify the calculation, we perform a change of variables: $\boldsymbol{y} = \boldsymbol{\Sigma}^{-1/2}(\boldsymbol{x} - \boldsymbol{\mu})$. Under this transformation: $(\boldsymbol{x} - \boldsymbol{\mu})^T \boldsymbol{\Sigma}^{-1}(\boldsymbol{x} - \boldsymbol{\mu}) = \boldsymbol{y}^T \boldsymbol{y} = \|\boldsymbol{y}\|^2$, and the differential $d\boldsymbol{x}$ transforms as: $d\boldsymbol{x} = |\boldsymbol{\Sigma}^{1/2}| \, d\boldsymbol{y} = |\boldsymbol{\Sigma}|^{1/2} \, d\boldsymbol{y}$. Substituting these into the integral, we get:

$$\int_{\mathbb{R}^d} \exp\left(-(\boldsymbol{x} - \boldsymbol{\mu})^T \boldsymbol{\Sigma}^{-1}(\boldsymbol{x} - \boldsymbol{\mu})\right) d\boldsymbol{x} = |\boldsymbol{\Sigma}|^{1/2} \int_{\mathbb{R}^d} \exp(-\|\boldsymbol{y}\|^2) \, d\boldsymbol{y} = \pi^{n/2}, \tag{31}$$

since the remaining integral is a standard Gaussian integral. Thus, the $L^2$-norm integral becomes:

$$\int_{\mathbb{R}^d} p(\boldsymbol{x})^2 \, d\boldsymbol{x} = \frac{1}{(2\pi)^n |\boldsymbol{\Sigma}|} |\boldsymbol{\Sigma}|^{1/2} \pi^{n/2} = \frac{1}{2^n \pi^{n/2} |\boldsymbol{\Sigma}|^{1/2}}. \tag{32}$$

Since the integral is a finite constant, we conclude that the PDF belongs to $L^2(\mathbb{R}^d)$.

$\square$

**Corollary 1** *Given an Euclidean space $D \subseteq \mathbb{R}^d$, a finite sum of Dirac deltas can be approximated with a mixture of multivariate Gaussian distributions:*

$$p_M(\boldsymbol{x}) = \sum_{i=1}^{n} w_i \cdot \mathcal{N}(\boldsymbol{x}; \boldsymbol{\mu_i}, \boldsymbol{\Sigma}_i). \tag{33}$$

**Proof.** Note that we do not show that a mixture of multivariate Gaussian distributions is the best candidate to approximate a finite sum of Dirac deltas. However, note that a multivariate Gaussian distribution is a standard approximation of a Dirac delta function and can, in the limit of a small covariance matrix, i.e. $|\boldsymbol{\Sigma}| << 1$, approximate it.

The aim of this proof is to show that the mixture of Gaussians $p_M$ can be a candidate to approximate the Dirac deltas. From Theorem 1, this is equivalent to showing that $p_M$ is a $L^2$ function.

To prove this, we need to integrate:

$$\int_D p_M(\boldsymbol{x})^2 d\boldsymbol{x} = \sum_{i=1}^{k} w_i^2 \int_D \mathcal{N}(\boldsymbol{x}; \boldsymbol{\mu}_i, \boldsymbol{\Sigma}_i)^2 d\boldsymbol{x} + 2 \sum_{i \neq j} w_i w_j \int_D \mathcal{N}(\boldsymbol{x}; \boldsymbol{\mu}_i, \boldsymbol{\Sigma}_i) \mathcal{N}(\boldsymbol{x}; \boldsymbol{\mu}_j, \boldsymbol{\Sigma}_j) d\boldsymbol{x}. \tag{34}$$

On the one hand, Lemma 1 shows that the integrals of the first sum are finite constants. On the other hand, the integrals on the second sum cannot be computed in a closed form, but it is well known that the product decays exponentially as $\|\boldsymbol{x}\| \to \infty$ ensuring a finite integral. Therefore, the squared integral is just a sum of finite constants and hence finite. $\square$

## A.3 Sampling Algorithm

---

**Algorithm 2** Sampling

---
1: $X_T \sim \lambda_\epsilon$
2: **for** $t = T, \ldots, 1$ **do**
3: $\quad \widetilde{X}_t^{thin} \sim g_\theta(\boldsymbol{x} \in X_t^{thin} | X_t, t)$
4: $\quad \widetilde{X}_0 \setminus X_t \sim f_\theta(X | X_t, t)$
5: $\quad \widetilde{X}_0 = (\widetilde{X}_0 \setminus X_t) \cup \widetilde{X}_t^{thin}$
6: $\quad X_{t-1} \sim q(X_{t-1} | \widetilde{X}_0, X_t)$
7: **end for**
8: **return** $X_{t-1}$

---

## A.4 Model Setup

**Architecture:** The classifier to predict $X_0 \cap X_t$ is a MLP with 3 layers and ReLU as activation function. The mixture of multivariate Gaussian distribution that approximates $X_0 \setminus X_t$ contains 16 components, and the parameters are learned with an MLP of 2 layers and ReLU as an activation function.

**Training:** All models have been trained on an NVIDIA A100-PCIE-40GB. We use *Adam* as the optimizer and a fixed weight decay of $0.0001$ to avoid overfitting. To avoid exploding gradients, we clip the gradients to have a norm lower than 2.

**Hyperparameters:** We use the same hyperparameters for all datasets and types of point processes. In a hyperparameter study A.8, we have found $T = 100$ for our cosine noise schedule (Nichol et al., 2021) to give a good trade off between sampling time and quality. Further, we leverage a hidden dimension and embedding size of 32. For training, we use a batch size of 128 and a learning rate of $0.001$.

**Early stopping:** We train the models up to 5000 epochs with early stopping, sampling 100 sequences from the model and comparing them to the validation split, with WD-SL metric for SPP and the CD-MMD metric for STPPs.

## A.5 Experimental results with standard deviations

Table 5: Density estimation results on the hold-out test set for SPPs averaged over three random seeds.

| | Earthquakes | | Covid NJ | | Citybike | | Pinwheel | |
|---|---|---|---|---|---|---|---|---|
| | SL($\downarrow$) | MMD($\downarrow$) | SL($\downarrow$) | MMD($\downarrow$) | SL($\downarrow$) | MMD($\downarrow$) | SL($\downarrow$) | MMD($\downarrow$) |
| Log-Gaussian Cox | $0.047 \pm 0.014$ | $0.214 \pm 0.004$ | $0.209 \pm 0.011$ | $0.340 \pm 0.008$ | $0.104 \pm 0.017$ | $0.336 \pm 0.014$ | $0.017 \pm 0.004$ | $0.285 \pm 0.004$ |
| Regularized Method | $2.361 \pm 0.064$ | $0.391 \pm 0.004$ | $0.255 \pm 0.011$ | $0.411 \pm 0.003$ | $0.097 \pm 0.008$ | $0.342 \pm 0.008$ | $0.039 \pm 0.003$ | $0.411 \pm 0.004$ |
| POINT SET DIFFUSION | $0.038 \pm 0.003$ | $0.173 \pm 0.004$ | $0.199 \pm 0.002$ | $0.268 \pm 0.016$ | $0.056 \pm 0.020$ | $0.092 \pm 0.020$ | $0.017 \pm 0.003$ | $0.099 \pm 0.006$ |

Table 6: Conditional generation results on the hold-out test set for SPP averaged over three random seeds.

| | Earthquakes | | Covid NJ | | Citybike | | Pinwheel | |
|---|---|---|---|---|---|---|---|---|
| | MAE($\downarrow$) | WD($\downarrow$) | MAE($\downarrow$) | WD($\downarrow$) | MAE($\downarrow$) | WD($\downarrow$) | MAE($\downarrow$) | WD($\downarrow$) |
| Regularized Method | $30.419 \pm 0.278$ | $0.162 \pm 0.003$ | $16.075 \pm 0.236$ | $0.148 \pm 0.001$ | $7.740 \pm 0.173$ | $0.115 \pm 0.001$ | $3.547 \pm 0.104$ | $0.150 \pm 0.003$ |
| POINT SET DIFFUSION | $4.651 \pm 0.159$ | $0.106 \pm 0.001$ | $5.056 \pm 0.115$ | $0.119 \pm 0.001$ | $3.498 \pm 0.365$ | $0.085 \pm 0.014$ | $2.256 \pm 0.037$ | $0.122 \pm 0.001$ |

Table 7: Density estimation results on the hold-out test set for STPP averaged over three random seeds.

| | Earthquakes | | Covid NJ | | Citybike | | Pinwheel | |
|---|---|---|---|---|---|---|---|---|
| | SL($\downarrow$) | MMD($\downarrow$) | SL($\downarrow$) | MMD($\downarrow$) | SL($\downarrow$) | MMD($\downarrow$) | SL($\downarrow$) | MMD($\downarrow$) |
| DeepSTPP | 0.105± 0.027 | 0.266± 0.041 | 0.169± 0.089 | 0.166± 0.177 | 3.257± 0.685 | 0.677± 0.056 | 1.067± 0.893 | 0.197± 0.152 |
| DiffSTPP | 0.088± 0.009 | 0.064± 0.024 | 0.332± 0.012 | 0.146± 0.026 | 0.560± 0.045 | 0.611± 0.113 | 0.196± 0.098 | 0.055± 0.005 |
| AutoSTPP | 0.073± 0.007 | 0.062± 0.004 | 0.364± 0.040 | 0.280± 0.202 | 0.598± 0.047 | 0.331± 0.099 | 0.127± 0.004 | 0.147± 0.005 |
| POINT SET DIFFUSION | 0.042± 0.003 | 0.023± 0.003 | 0.189± 0.006 | 0.043± 0.003 | 0.032± 0.004 | 0.020± 0.001 | 0.023± 0.003 | 0.020± 0.001 |

Table 8: Forecasting results on the hold-out test set for STPP averaged over three random seeds.

| | Earthquakes | | Covid NJ | | Citybike | | Pinwheel | |
|---|---|---|---|---|---|---|---|---|
| | MAE($\downarrow$) | CD($\downarrow$) | MAE($\downarrow$) | CD($\downarrow$) | MAE($\downarrow$) | CD($\downarrow$) | MAE($\downarrow$) | CD($\downarrow$) |
| DeepSTPP | 10.154 ± 0.918 | 11.211 ± 0.738 | 6.264 ± 0.378 | 8.492 ± 0.196 | 127.968 ± 33.298 | 125.747 ± 32.705 | 18.651 ± 7.159 | 15.792 ± 5.323 |
| DiffSTPP | 16.027 ±6.833 | 17.466 ±5.748 | 18.822 ±3.381 | 14.302 ±0.216 | 7.516 ±1.973 | 8.460 ±1.773 | 14.461 ±4.816 | 13.062 ±3.901 |
| POINT SET DIFFUSION | 7.407 ±0.285 | 10.458 ±0.218 | 7.293 ±0.082 | 10.865 ±0.130 | 5.928 ±2.881 | 7.225 ±2.802 | 6.341 ±0.108 | 6.437 ±0.124 |

## A.6 PERFORMANCE COMPARISON TO ADD-THIN ON THEIR TPP EXPERIMENTS

We compare our POINT SET DIFFUSION to ADD-THIN (Lüdke et al., 2023) on their TPP experiments. We use the same training and hyper-parameter setup for our model as in the SPP and STPP experiments. For details on the experimental setup, please refer to section 5 of Lüdke et al. (2023).

### A.6.1 DENSITY ESTIMATION

Table 9: MMD ($\downarrow$) between the TPP distribution of sampled sequences and hold-out test set (**bold** best).

| | Hawkes1 | Hawkes2 | SC | IPP | RP | MRP | PUBG | Reddit-C | Reddit-S | Taxi | Twitter | Yelp1 | Yelp2 |
|---|---|---|---|---|---|---|---|---|---|---|---|---|---|
| ADD-THIN | **0.02** | **0.02** | **0.19** | 0.03 | **0.02** | 0.10 | **0.03** | **0.01** | **0.02** | **0.04** | **0.04** | 0.08 | **0.04** |
| POINT SET DIFFUSION | 0.03 | 0.03 | **0.19** | **0.02** | 0.04 | **0.07** | 0.05 | **0.01** | **0.02** | 0.11 | 0.09 | **0.06** | 0.06 |

Table 10: Wasserstein distance ($\downarrow$) between the distribution of the number of events of sampled sequences and hold-out test set (**bold** best).

| | Hawkes1 | Hawkes2 | SC | IPP | RP | MRP | PUBG | Reddit-C | Reddit-S | Taxi | Twitter | Yelp1 | Yelp2 |
|---|---|---|---|---|---|---|---|---|---|---|---|---|---|
| ADD-THIN | 0.04 | **0.02** | 0.08 | **0.01** | **0.02** | 0.04 | 0.02 | **0.03** | **0.04** | **0.03** | **0.01** | 0.04 | **0.02** |
| POINT SET DIFFUSION | **0.03** | 0.03 | **0.03** | **0.01** | 0.03 | **0.02** | **0.01** | **0.03** | 0.03 | 0.10 | **0.01** | **0.03** | 0.03 |

### A.6.2 CONDITIONAL GENERATION – FORECASTING

Table 11: Wasserstein distance ($\downarrow$) between forecasted event sequence and ground truth reported for 50 random forecast windows on the test set (lower is better).

| | PUBG | Reddit-C | Reddit-S | Taxi | Twitter | Yelp1 | Yelp2 |
|---|---|---|---|---|---|---|---|
| Average Seq. Length | 76.5 | 295.7 | 1129.0 | 98.4 | 14.9 | 30.5 | 55.2 |
| ADD-THIN | 2.03 | 17.18 | 21.32 | **2.42** | **1.48** | 1.00 | 1.54 |
| POINT SET DIFFUSION | **1.98** | **16.90** | **16.23** | 2.52 | 1.51 | **0.96** | **1.50** |

Table 12: Count MAPE $\times 100\%$ ($\downarrow$) between forecasted event sequences and ground truth reported for 50 random forecast windows on the test set (lower is better).

| | PUBG | Reddit-C | Reddit-S | Taxi | Twitter | Yelp1 | Yelp2 |
|---|---|---|---|---|---|---|---|
| Average Seq. Length | 76.5 | 295.7 | 1129.0 | 98.4 | 14.9 | 30.5 | 55.2 |
| ADD-THIN | 0.45 | **1.07** | 0.38 | **0.37** | 0.69 | **0.45** | 0.50 |
| POINT SET DIFFUSION | **0.44** | 1.13 | **0.26** | 0.41 | **0.60** | 0.46 | **0.47** |

A.7 ADDITIONAL MATERIAL FOR COMPUTATIONAL COMPLEXITY OF STPP MODELS

Table 13: Number of learnable parameters per model.

| DEEPSTPP | DIFFSTPP | AUTOSTPP | POINT SET DIFFUSION |
|---|---|---|---|
| $\sim 450,000$ | $\sim 1,600,000$ | $\sim 1,000,000$ | $\sim 25,000$ |

Table 14: Training runtime in minutes averaged over three random seeds (all models have been trained on an A100).

| | Earthquakes | Covid NJ | Citybike | Pinwheel |
|---|---|---|---|---|
| DEEPSTPP | 32 | 28 | 71 | 17 |
| DIFFSTPP | 469 | 492 | 832 | 392 |
| AUTOSTPP | 99 | 156 | 523 | 36 |
| POINT SET DIFFUSION | 52 | 42 | 68 | 50 |

A.8 HYPERPARAMETER STUDY $T$

To provide insight into how the number of steps affects sample quality, we have run a hyperparameter study for the unconditional STPP experiment on the validation set of the Earthquake dataset, evaluating $T \in \{20, 50, 100, 200\}$, averaged over three random seeds. Our findings indicate that while fewer diffusion steps result in reduced sample quality, $T = 100$ strikes a good balance, already matching and even surpassing the quality observed at $T = 200$. Although this result may seem counterintuitive to those familiar with standard Gaussian diffusion models, it highlights a key distinction of our approach: unlike Gaussian diffusion processes, our model employs inherently discrete Markov steps—specifically, the superposition and thinning of point sets with fixed cardinality. As a result, only a limited number of points can be added or removed over $T$ steps, imposing a natural ceiling on how much additional steps can improve sample quality.

Table 15: STPP density estimation results on the Earthquake validation set for $T \in \{20, 50, 100, 200\}$ reported as the average and standard error over three random seeds.

| | 20 | 50 | 100 | 200 |
|---|---|---|---|---|
| SL ($\downarrow$) | $0.018 \pm 0.002$ | $0.017 \pm 0.002$ | $0.014 \pm 0.002$ | $0.015 \pm 0.001$ |
| MMD ($\downarrow$) | $0.020 \pm 0.0015$ | $0.020 \pm 0.0002$ | $0.018 \pm 0.0012$ | $0.018 \pm 0.0005$ |

A.9 STPP FORECASTING DENSITY EVOLUTION

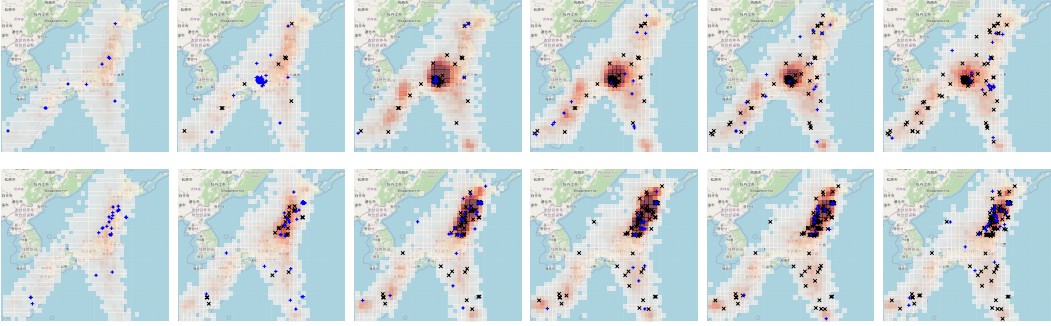

Figure 8: Evolution of two STPP forecasts of POINT SET DIFFUSION across time ($0 \rightarrow t_{max}$): Density plot of forecast for a sliding window of $\frac{1}{6}$ of the maximum time, black crosses represent history (conditioning), blue ground-truth future.

