# OpenReview forum: "Unlocking Point Processes through Point Set Diffusion"
_ICLR.cc/2025/Conference — ICLR 2025 Poster_

### Official Review · Reviewer_8FLx · 2024-11-01

**Soundness:** 3
**Presentation:** 3
**Contribution:** 3
**Rating:** 6
**Confidence:** 3

**Summary:**

This paper proposes a novel diffusion-based approach to model point processes without relying on traditional intensity functions. This model is characterized by its ability to efficiently and flexibly generate point sets through stochastic interpolation between data and noise sets. Experiments on synthetic and real-world datasets demonstrate that the model achieves state-of-the-art performance in generating spatial and spatiotemporal point processes, significantly outperforming existing methods in terms of speed and accuracy of sample generation.

**Strengths:**

- The approach of modeling point processes using the diffusion model is interesting.
- Efficient sampling is achieved by making effective use of thinning.
- The effectiveness of the proposed method is evaluated on artificial and real data.
- The manuscript is well-written.

**Weaknesses:**

- There is not enough discussion about computational complexity.
- Not very clear on how to set hyperparameters.
- No mention of the effectiveness of the method with respect to the amount of data.
- No discussion of limitation.

**Questions:**

- Can you add a discussion on learning time? How does the computational complexity increase, especially with more data points?
- How can hyperparameters (e.g. number of diffusion steps T or noise scheduling) be determined?
- Please tell me more about the limitation of the proposed method. For example, how robust is the proposed method in situations where there is little data? Also, will the interpretability of the proposed method be lower than parametric methods (e.g., DNN-based Hawkes processes), or will the number of sensitive hyperparameters increase by using diffusion models as a base?

---

> ### Author Response · Authors · 2024-11-22
>
> # Response
>
> We would like to thank the reviewer for their thoughtful and constructive feedback.
> We appreciate the points raised and hope the following addresses the questions and suggestions.
>
> ### W1, Q1 & W3: Computational complexity
> We have added the average training runtimes to the appendix (see A.8), demonstrating that our model is comparatively fast to train.
> Notably, our model has the fewest learnable parameters—up to two orders of magnitude fewer than the baselines—further allowing to reduce the effective compute time by running multiple models in parallel on a single GPU.
> Furthermore, as shown in Figure 6 and discussed in Section 4.4, the computational complexity with respect to the number of points in a set remains nearly constant for our model.
> While the full attention in our encoder could eventually impact scaling, this could then be mitigated by limiting the attention to a fixed context window, a practice already leveraged by the baselines to keep them tractable.
>
> ### W2 & Q2: Hyperparameter selection and sensitivity
> We generally find our model to be robust across different hyperparameter values, allowing us to use the same hyperparameters for all datasets. We added the results for different numbers of diffusion steps $T$ to the appendix (see A.9) and have discussed them in our response to **reviewer r6U8**; for further discussion of the noise schedule, please also refer to our response to **reviewer mmd5**.
>
> ### Q3: Interpretability
> Without ordering, it is generally not possible to effectively model or sample the conditional Janossy intensity (Eq. 2) for point processes on general metric spaces.
> To address this, our proposed method learns the joint density of point sets through a diffusion-based latent variable model.
> While this enables modeling point processes on general metric spaces, supports efficient parallel sampling, and allows for flexible generation in complex conditional tasks, it does not permit evaluating or interpreting the conditional intensity or its parameters.
> Therefore, for applications involving ordered point processes (STPP, TPP) that require evaluation of the conditional intensity—such as estimating the likelihood of the next point given the past—point process models that directly approximate the conditional intensity are better suited.
> In contrast, our method prioritizes generality, scalability, and flexibility, excelling in unconditional and conditional generation tasks and addressing key limitations of intensity-based models.
> We have added a small discussion of this limitation to the last paragraph of our conclusion.

---

### Official Review · Reviewer_mmd5 · 2024-11-02

**Soundness:** 3
**Presentation:** 3
**Contribution:** 3
**Rating:** 8
**Confidence:** 4

**Summary:**

This paper proposes a diffusion-based latent variable model for general point processes on metric spaces. By learning stochastic interpolations between data and noise point sets, the model enables efficient, parallel sampling and flexible generation for complex tasks on the metric space. Experiments on synthetic and real-world datasets show that the proposed model achieves state-of-the-art results in unconditional and conditional tasks for spatial and spatio-temporal point processes.

**Strengths:**

1. This paper generalizes the Add-Thin model to define a model for point processes on general metric spaces, enhancing the model's applicability and promising future prospects.
2. The idea is sound and well-founded. The paper is overall well-written and easy to follow.
3. Experiments show that the proposed model achieves state-of-the-art results on both conditional and unconditional tasks while enabling faster sampling.

**Weaknesses:**

1. It would be helpful to discuss the connections between the proposed model and the Add-Thin model when modeling univariate temporal point processes.
2. In the conditional sampling, the definition of $q(X_{t-1} | X_{0}^c)$ in line 287 was not provided.

Typo: $X_{t+1}^{\text{thin}}$ and $X_{t}^{\text{thin}}$ in Eq.(9) should be $X_{t+1}^{\varepsilon}$ and $X_{t}^{\varepsilon}$.

**Questions:**

1. In the experiments, how are $\alpha_t$, $\beta_t$, and $T$ set?

2. The proposed model generalizes the Add-Thin model to general metric spaces. When modeling univariate TPPs, how does the performance of the proposed model compare to Add-Thin in both unconditional and conditional sampling scenarios?

**Details Of Ethics Concerns:**

None.

---

> ### Author Response · Authors · 2024-11-22
>
> # Response
>
> We appreciate the reviewer’s detailed evaluation and feedback on our work and respond to the comments below.
>
> ### W2: Definition of $q(X_{t-1}|X_0^c)$
> $q(X_{t-1}|X_0^c)$ in Algorithm 1 refers to the Markov Chain of the forward process (Section 3.1), which intuitively noises the condition by thinning and adding noise on the whole domain.
> Then by applying the conditioning mask (line 6), we obtain the noised condition.
> We agree that the brevity of this notation might be hard to follow, and we annotated the lines in the algorithm accordingly.
>
>
> ### Typos
> Thanks for pointing out the typos; we have adjusted the manuscript accordingly.
>
> ### W1 & Q2: Connection and performance difference to ADD-THIN
> First, ADD-THIN also leverages the thinning and superposition properties to define a diffusion process for TPPs, where they mix the thinning and superposition in their noising process so that it consists of the superposition of $T+1$ point sets with different intensity functions, where even added points can be removed again, which significantly complicates the posterior and introduces redundant steps.
> In contrast, our model disentangles the superposition and thinning to attain two independent processes to allow for more explicit control and define the diffusion model independent of the intensity function as a stochastic interpolation of two point sets.
> Further, the parametrization of ADD-THIN is specific to TPPs and directly leverages the ordering of points (temporal embeddings, inter-event times, convolutional layers), while POINT SET DIFFUSION is agnostic to the ordering of points, making it applicable for modeling the general class of point processes on any metric space, including for example SPPs.
> Lastly, ADD-THIN needs to be explicitly trained for specific conditioning tasks, while we show how, after training, our unconditional POINT SET DIFFUSION model can be conditioned for arbitrary and unknown conditioning tasks on the metric space.
> We discuss these connections in our related work section in more generality.
>
> Second, to show the performance difference, we ran all TPP experiments from ADD-THIN with our model and report them in appendix A7.
> As can be seen, we are able to match the SOTA performance of ADD-THIN on TPPs with our more general setting.
> It is especially noteworthy that our unconditionally trained model can surpass the conditionally trained ADD-THIN model on their forecasting task, which shows that our model can effectively capture the interaction between points by directly modeling the joint density.
>
> ### Q1: Noise schedule
> $T$ is set to 100 steps, where we refer the reviewer to our discussion of the impact of $T$ in our answer to **reviewer r6U8**.
> Regarding $\alpha$ and $\beta$, we have found $\bar{\alpha}_t = 1-\bar{\beta}_t$ to be effective since it ensures a direct interpolation between the two point sets, which if $\int_A \lambda^{\epsilon} = E[N(A)]$ ensures a constant expected number of points throughout the process.
> Regarding the choice of a noise schedule, we initially experimented with a linear and a cosine schedule, where we found the cosine schedule to work marginally better.
> However, the specifics of the noise schedule are an interesting direction for future work to explore, especially since most (continuous) noise schedules were designed for continuous Gaussian diffusion models focusing on the noise-to-signal ratio of a continuous variable.
> In contrast, our diffusion process is distinctively more discrete (i.e., thinning and superposition process), possibly opening a new pathway for future work.

---

> > ### Comment · Reviewer_mmd5 · 2024-11-23
> > **Response**
> >
> > Thank you, the response has completely addressed my questions, and I have improved my score to 8.

---

### Official Review · Reviewer_r6U8 · 2024-11-03

**Soundness:** 3
**Presentation:** 4
**Contribution:** 3
**Rating:** 8
**Confidence:** 3

**Summary:**

The paper proposes a Point Set Diffusion model for conditioned and unconditioned generations of point processes (spatial, temporal, and spacial-temporal) without intensity functions. The model treats the latent space of the point process as a whole and applies diffusion to learn how to generate point processes from noise (unconditioned) and conditioning masks (conditioned). At the training phase, the point process is passed through a forward process that gradually thins the original points and adds points from a noise point process. Then, a parameterized model is trained for the backward process that gradually predicts the points in the last timestep conditioned on the current timestep and thins the noise points in the current point process. After training, both conditioned and unconditioned sampling procedures are provided. Numerical experiments illustrate that the proposed Point Set Diffusion model achieves much faster sampling speed than intensity-based autoregressive models. Moreover, it outperforms several baseline autoregressive models on various SPP and STPP tasks, especially density estimation tasks.

**Strengths:**

1. The paper is overall well-written and easy to follow. The basic concepts are introduced clearly with consistent notations. The forward process, backward process, and final sampling algorithms are well explained. Illustrations (Figure 1-3) are very clear for readers to follow the workflow of the proposed Point Set Diffusion model. The datasets and metrics are also clear in the experiment section.

2. The idea of leveraging diffusion models to generate the whole point process is intriguing, and it is quite different from the common approaches that use autoregressive models with parameterized intensity functions that suffer from sampling speed and are restricted to forecasting tasks. Numerical results are very promising to support the efficacy of the proposed model.

**Weaknesses:**

1. Currently there are very few baseline algorithms, e.g., for SPP conditional generation there is only one baseline, and for STPP forecasting there are only two. It would be more convincing to compare with more baseline models, or to provide more evidence that the current baselines are already SOTA (which I believe they are).

**Questions:**

1. The sampling time and quality of the diffusion model are directly related to the number of forward/backward steps, which I could not find in the paper. Could the authors provide some ablation study on the number of steps, e.g., how the sampling time and quality grow with it?

---

> ### Author Response · Authors · 2024-11-22
>
> # Response
>
> We want to thank the reviewer for their feedback and appreciation of our work and address their two points.
>
> ### W1: SPP baselines
> We agree that while the very popular (Log-Gaussian) Cox process and the Regularized Model proposed at NeurIPS 2019 are highly relevant baselines, comparisons to additional SPP baselines would be valuable.
> However, the unordered nature of SPPs does not permit direct and effective modeling or sampling of the conditional Janossy intensity, thereby precluding the application of traditional point process models designed for ordered spaces.
> To the best of our knowledge, there exists no other SPP model capable of capturing point-to-point interactions that underlie the unconditional or conditional SPP tasks.
> We hope this explanation clarifies our choice of baselines and emphasizes the challenges in identifying comparable methods within the scope of this work.
>
>
> ### Q1: Trade-off between sampling time and quality $(T)$
>
> | $T$ (Diffusion Steps) | SL (avg, Standard Error) | MMD (avg, Standard Error)|
> |---------------------------|-----------------|-----------------|
> | 20                        | 0.018 ± 0.002     | 0.020 ± 0.0015    |
> | 50                        | 0.017 ± 0.002     | 0.020 ± 0.0002     |
> | 100                       | 0.014 ± 0.002     | 0.018 ± 0.0012     |
> | 200                       | 0.015 ± 0.001     | 0.018 ± 0.0005     |
>
> We have used $T=100$ for all experiments and added a sentence to the hyperparameter paragraph in the Appendix.
>
> The sampling time scales linearly with the number of diffusion steps $T$.
> To provide insight into how the number of steps affects sample quality, we have run a hyperparameter study for the unconditional STPP experiment on the validation set of the Earthquake dataset, evaluating $T \in \{ 20, 50, 100, 200 \}$, averaged over three random seeds.
> Our findings indicate that while fewer diffusion steps result in reduced sample quality, $T = 100$ strikes a good balance, already matching and even surpassing the quality observed at $T = 200$.
> Although this result may seem counterintuitive to those familiar with standard Gaussian diffusion models, it highlights a key distinction of our approach: unlike Gaussian diffusion processes, our model employs inherently discrete Markov steps—specifically, the superposition and thinning of point sets with fixed cardinality.
> As a result, only a limited number of points can be added or removed over $T$ steps, imposing a natural ceiling on how much additional steps can improve sample quality.

---

> > ### Comment · Reviewer_r6U8 · 2024-11-23
> >
> > Thank you, the response addresses my question and concern. I keep my score and recommend for acceptance.

---

### Official Review · Reviewer_XRB4 · 2024-11-04

**Soundness:** 3
**Presentation:** 4
**Contribution:** 2
**Rating:** 6
**Confidence:** 5

**Summary:**

This paper proposes a novel modeling approach to point processes via diffusion on point sets (discrete events), addressing the reliance on the intensity function when establishing or learning the model. It can capture the distribution of point processes and generate a series of events based on noise point sets. Meanwhile, the sampling efficiency of point set diffusion is superior. The overall presentations of both the methodology and experiments are excellent.

**Strengths:**

The idea of using the diffusion-style model to characterize point processes is super interesting. The content is clear and well-written, making the methodology and the results accessible to the reader. The paper also covers unconditional and conditional sampling methods, which have the potential to correspond to two important questions in the point process modeling (first-order and second-order modeling). The authors also provide thorough experimentation to validate the effectiveness of the proposed model.

**Weaknesses:**

In my opinion, the main weakness, or the most improvement-needed part of the paper, lies in the modeling and experiments of ordered point processes:

1. An important characteristic of the ordered point processes (TPPs or STPPs) is the dependence between future events and past events, which is not considered in the model. The proposed method seems to only consider the first-order statistics of the data (the event intensity/density), and treat these statistics at certain times or locations as fixed values to be learned by the model. For example, if the training data set contains multiple event trajectories sampled over the horizon of $[0, T]$, then the model will assume $p(T/2)$ (the event density at $T/2$) is fixed and to be learned. However, in TPPs, the $p(T/2)$ depends on the history (observation before $T/2$), and is different in each realization of the event trajectory, which violates the assumption of the diffusion model.

2. Although a conditional sampling method is proposed in the paper (Algorithm 3.1), I am wondering about its effectiveness in practice. First, what the $q(X_{t-1}|X_{0}^{c})$ is (line 287) remains unknown. Meanwhile, the (technical/practical) reason for using this conditioning is not shown. The results in Figure 7 are not convincing enough. To me, even though the authors claim that they are solving conditioning tasks and are visualizing the predicted densities for events from different trajectories (panels at the bottom), these density plots would look similar if we overlap them with each other. In other words, I think the model only predicts an averaged event intensity over space, and it has little connection with the conditioned samples.

3. An alternative to prove the effectiveness of conditional generation is to show the predicted intensity/density of events at different times, given a trajectory from the pinwheel dataset. This is the same idea as Figure 5 in [1]. The difference between density functions at different times is more significant and would help validate the conditional sampling method.

4. The conditional sampling task is only experimented with in the spatial domain. An example of showing the evolution of the predicted conditional density of a pinwheel trajectory can support the claim of effective conditional sampling in an ordered (temporal) domain.

5. I am also concerned that there is no log-likelihood metric reported in the paper. The metrics used in the paper are about the first-order characteristics of the data, on which I believe the proposed Point Set Diffusion can perform well. However, they cannot fully reflect the model's fit to the data when second-order data dependencies are involved (e.g., in ordered point processes). On the other hand, the log-likliehood is still the golden standard to suggest the model's goodness-of-fit to the data when it comes to conditional models or tasks [2][3]. Other point process studies that use the diffusion model will also report the data log-likelihood when evaluating the model [4][5]. I am curious about the proposed model's performance on the log-likelihood metric.

Again, I acknowledge and respect the authors' contribution to the proposed method, and I hope the above questions can be properly answered or addressed.

---
[1] Chen, Ricky TQ, Brandon Amos, and Maximilian Nickel. "Neural Spatio-Temporal Point Processes." International Conference on Learning Representations.

[2] Daryl J Daley, David Vere-Jones, et al. An introduction to the theory of point processes: volume I: elementary theory and methods. Springer, 2003

[3] Reinhart, Alex. "A review of self-exciting spatio-temporal point processes and their applications." Statistical Science 33.3 (2018): 299-318.

[4] Dong, Zheng, Zekai Fan, and Shixiang Zhu. "Conditional Generative Modeling for High-dimensional Marked Temporal Point Processes." arXiv preprint arXiv:2305.12569 (2023).

[5] Yuan, Yuan, et al. "Spatio-temporal diffusion point processes." Proceedings of the 29th ACM SIGKDD Conference on Knowledge Discovery and Data Mining. 2023.

**Questions:**

1. In Figure 5, can the authors show the predicted density of different trajectories in the same masked area?

---

> ### Author Response · Authors · 2024-11-22
>
> # Response
>
> We thank the reviewer for their appreciation of our work and their feedback.
>
> As the raised points seem rooted in one misunderstanding of our model, we clarify it before addressing the specific comments.
>
> ## Our model captures the joint density of point sets
> When modeling Point Processes, we are interested in capturing the complex interactions between all points, which can be expressed as the **conditional** Janossy intensity (see Eq. 2).
> However, as explained in the subsequent paragraph, parameterizing and sampling this intensity is not feasible in general for non-ordered spaces.
> For ordered point processes (TPP, STPP), most models rely on the history-dependent intensity to parameterize the density of the point process, i.e., $p(X)= \prod^N_i \lambda(x_i|H_{x_i}) e^{-\int \lambda(x|H_{x})}$, where $H_{x}$ is the conditional history up to point $x$.
> This introduces a factorization across time, enabling autoregressive sampling but limiting conditioning tasks to simple forecasting and enforcing sequential sampling.
> In contrast, our approach leverages the thinning and superposition properties to sample from any conditional Janossy intensity without explicitly parameterizing it.
> Intuitively, one can think of it as a different factorization of the Point Process density $p(X)$, not across time, but across a latent variable process -- Point Set Diffusion.
> Thus, our model is neither restricted to inhomogeneous Point Processes, i.e., $\lambda(x_i|H_{x_i})= \lambda(x_i)$ nor first-order statistics but generalizes to point processes with arbitrary interactions.
>
>
> ### W1: Point set diffusion is not limited to "first-order-statistics"
> As noted above, our model is not limited to first-order statistics or inhomogeneous intensities. It generalizes beyond conditional intensities of ordered processes, capturing any interaction between points on the metric space. For instance, it can predict the history given the future or a time window given past and future points.
> In short, our model supports a broader range of intensity functions than the conditional intensities captured by standard STPP or TPP models.
>
> ### W2: Why use the conditional sampling method (Algorithm 3.1)
> Our model is trained unconditionally to learn the joint density of point sets, so it is non-trivial to solve conditioning tasks by leveraging our joint density parametrization.
> With Algorithm 3.1, we demonstrate the flexibility of our model and show how to condition our unconditionally-trained model for conditioning tasks on the metric space, subsuming forecasting, history prediction, and more complex conditioning tasks.
> This algorithm is used for all conditional tasks in the paper: spatial conditioning (Sections 4.3, 4.5), temporal forecasting (Section 4.4, Appendix A.7.2, A.10).
>
>
> #### Clarifying definition of $q(X_{t-1}|X_0^c)$
> $q(X_{t-1}|X_0^c)$ refers to the Markov Chain of the forward process (Sec. 3.1), which noises the condition by thinning and adding noise across the domain.
> Applying the conditioning mask (Algorithm 3.1, line 6) yields the noised condition. We agree that this notation may be unclear and have annotated the algorithm for clarity.
>
>
>
> ### W2, W3, W4: Conditional generation for (ordered) point processes
>
> #### Visualization
> While Figure 7’s density plots may appear similar, they are distinct. The spatial similarities are common for S(T)PPs, such as Earthquakes (Figure 7), and reflect a "shared" spatial pattern along tectonic plates.
> To better demonstrate our conditional generation, we added plots (Appendix A.10) showing the spatial density at different time points for the Earthquake dataset.
> Since, unlike other STPP or TPP models, our approach does not parameterize the intensity of the next point given the past but models all following points, we cannot show the conditional density plots requested in W3 and W4.
> However, in the plot, we show sliding forecast windows (e.g., forecasting 1/6th of the time domain at different time points, (0, 1/6, 2/6,...,5/6)), revealing that our conditional densities change in time and are influenced by prior events (e.g., earthquake aftershocks).
>
>
> #### Conditional results
> Sections 4.3 and 4.4 report state-of-the-art results for conditional tasks on SPPs and STPPs.
> Furthermore, Appendix A.7.2 compares our model to ADD-THIN [3] on their TPP forecasting task, where they recently showed state-of-the-art results.
> Note that our model, unlike ADD-THIN, is not specifically trained for this task.
> Still, our model closely matches or even outperforms ADD-THIN on all datasets.

---

> ### Author Response · Authors · 2024-11-22
>
> ### W5: Log-likelihood metric
> The log-likelihood (LL) measures the likelihood of the next event given its history, typically parameterized through the conditional intensity.
> Since our model does not parameterize this conditional intensity—unlike the diffusion models cited in the review—reporting the LL is not applicable.
>
> That said, we would like to respectfully challenge the notion of the LL as the gold standard for evaluation.
> The LL is computed as $\sum^N_i\lambda(x_i|H_{x_i}) - \int \lambda(x_i|H_{x_i})$, with the integral spanning space and time for STPPs.
> While the LL is a natural metric for STPP baselines trained to optimize it, it is evaluated using the ground truth history, primarily reflecting the next-event prediction but not the quality of generated samples or forecasts in real-world applications.
> Further, the LL has known failure modes, and models with high LL can still produce samples significantly different from the training data [5], an issue worsened by error accumulation in autoregressive sampling.
> This issue has been raised by different PP papers [1][2][3], with [1] even stating that "the NLL is mostly irrelevant as a measure of error in real-world applications."
> Similarly, probabilistic forecasting has replaced LL as the "gold standard" in time series [4][6].
> Additionally, the reported LL directly depends on model-specific approximations and parametrizations, complicating and making a fair comparison across (S)TPP models error-prone due to differing approximations and implementations.
>
> In contrast, our experiments follow [2] and [3] by comparing the distributions of point set samples for different tasks, independent of the implementation of each model.
> It is very important to state that the applied metrics capture more than just the first-order characteristics of the data.
> For the unconditional task, we report the Wasserstein distance between the count distributions and the maximum mean discrepancy.
> This kernel-based statistic test compares the two distributions based on a sample-based distance metric (CD) (for further details on the MMD for point processes, please also refer to Appendix E.2 of [2]).
> Given the ground truth target in the conditional tasks, we do not need to compare distributions of point processes.
> Hence, we leverage the MAE for the difference in the number of points and the Point Process Wasserstein Distance, again a distance between two distributions as each instance of a point process is a stochastic process.
>
> Lastly, restricting evaluations to one-step-ahead LL would limit model design to parameterizations of a conditional intensity function for which we can efficiently or approximately compute the integral–in our opinion, a significant restriction for the field of point processes.
>
>
> [1] Shchur, O., Türkmen, A. C., Januschowski, T., & Günnemann, S. "Neural temporal point processes: A review." IJCAI (2021)
>
> [2] Shchur, O., Gao, N., Biloš, M., & Günnemann, S. "Fast and flexible temporal point processes with triangular maps." NeurIPS (2020)
>
> [3] Lüdke, D., Biloš, M., Shchur, O., Lienen, M., & Günnemann, S. Add and thin: Diffusion for temporal point processes. NeurIPS, (2023)
>
> [4] Tilmann Gneiting and Matthias Katzfuss. Probabilistic forecasting. Annual Review of Statistics and Its Application, (2014)
>
> [5] Theis, Lucas, Aäron van den Oord, and Matthias Bethge. "A note on the evaluation of generative models." ICLR (2016)
>
> [6] Alexandrov et al. GluonTS: Probabilistic and neural time series modeling in Python. JMLR 2020

---

> ### Comment · Reviewer_XRB4 · 2024-11-27
>
> I appreciate the response from the authors. In general, some of my concerns have been addressed, but others have not, and I am not fully convinced in a few parts of the author's response.
>
> My confusion about "the model focuses on the first-order property of the point process" has been addressed, and I can agree with the logic behind the modeling of the **Janossy density**. I understand the necessity of a conditional sampling algorithm, and the newly added example of Appendix A.10 did demonstrate that future events' distributions are influenced by prior events. I give credit to the authors for helping address my above concerns.
>
> However, what I am concerned with in my previous review is the **effectiveness** or accuracy of the conditional sampling instead of the necessity. An easy way to prove this is via simulation: given an observed history generated from a true STPP model $m^*$, the authors can first use $m^*$ to generate enough samples of sequences in the future time frame and calculate the density of points as "distribution of future events". Then, the authors can use their model to generate future sequences and compare the distribution of those events with the true distribution of future events obtained from the $m^*$. A synthetic experiment like this can clearly prove whether the conditional sampling really captures the ground truth.
>
> #######
>
> The following are my comments on the discussion of the log-likelihood metric. With respect to the authors, I am afraid I cannot agree with their opinions on the log-likelihood metric:
> 1. First, the computation of the likelihood of a point process does not require the parametrization of the conditional intensity function. The intensity-based computation listed by the authors is only one of the approaches to compute the likelihood when the intensity function is available.
> 2. In fact, the derivation of the point process likelihood is closely connected with the **Janossy density** (see the derivation in section 2.4 in [1], or in Section 5.3/Definition 7.1.II in [2]). The likelihood is computed by a series of **density functions** (which is also the very original definition of any likelihood), seeing equation 6 in [1].
>     - A few examples of calculating likelihood without parametrizing the conditional intensity function can be found in [3][4]. These pp studies also use diffusion models to sample events (although I admit that they are using diffusion in an autoregressive way, and this paper contributes to it by extending the modeling beyond autoregression by sampling a few points in parallel), and the likelihood can be computed by sampling candidate points and calculating the density of observed ground truth.
>
> 3. If the authors are claiming that their model can learn the **Janossy density** of the point processes, I would also expect the model to have a good likelihood of the data. I can even think of a way for the authors to calculate the likelihood of a sequence: giving the observed history, generate multiple sequences, keep the first event in each sequence, and calculate the density. This is the density of the next event based on the history. The likelihood of an entire sequence is computed by iterating over all the events.
>     - I believe this evaluation procedure would cost linear complexity as others since the Point Set Diffusion is generating one sequence at a time.
>
> In summary, I hold my opinion about the golden standard of likelihood in point processes and cannot agree with the authors' opinion that it is the *significant restriction* for point processes. Again, it does not require the parameterization of the conditional intensity.
>
> #######
>
> Still, I understand that the computation of likelihood will not be a main flaw of the proposed Point Set Diffusion. I can now see the value of the proposed method to the fields of point processes. Meanwhile, I still hope the authors can consider adding the synthetic data experiments and the possible evaluation of the likelihood. This would improve the quality of the paper and make it look more convincing.
>
> Based on all the assessments above, I decided to raise my score.
>
> Do the authors plan to release the code/implementation?
>
> --------
> [1] Reinhart, Alex. "A review of self-exciting spatio-temporal point processes and their applications." Statistical Science 33.3 (2018): 299-318.
>
> [2] Daryl J Daley, David Vere-Jones, et al. An introduction to the theory of point processes: volume I: elementary theory and methods. Springer, 2003
>
> [3] Dong, Zheng, Zekai Fan, and Shixiang Zhu. "Conditional Generative Modeling for High-dimensional Marked Temporal Point Processes." arXiv preprint arXiv:2305.12569 (2023).
>
> [4] Yuan, Yuan, et al. "Spatio-temporal diffusion point processes." Proceedings of the 29th ACM SIGKDD Conference on Knowledge Discovery and Data Mining. 2023.

---

> > ### Author Response · Authors · 2024-12-02
> >
> > We thank the reviewer for their thoughtful response and are pleased that we resolved the confusion regarding our model's theoretical capacity.
> > We greatly appreciate the reviewer's updated assessment and recognition of our method's value and contribution, as reflected in the raised score.
> >
> >
> > #### Synthetic forecasting experiment
> > We believe our experiments on 17 real-world and synthetic datasets (SPPs: Sec. 4.3; STPPs: Sec. 4.4, 4.5, A.10; TPPs: A.7.2) demonstrate our model's effectiveness and accuracy on various conditioning tasks.
> > For (S)TPPs, these evaluate forecasting accuracy across over 50 forecast windows per test set instance.
> >
> > To complement them, we trained our model unconditionally on 1500 samples from a synthetic STPP Hawkes process (Hawkes1 setup in [1]) with a mixture of two exponential kernels and a Gaussian spatial diffusion kernel with constant variance.
> > This setup allows us to compute the likelihoods of entire point set samples w.r.t. the ground-truth process, as we know its conditional likelihood function.
> > The table below shows this negative log-likelihood (NLL)($\downarrow$) for 50 forecast samples on the last 5–40% for 200 Hawkes sequences, comparing our model to the ground-truth Hawkes and a misspecified Hawkes (same parameters, but homogeneous base-intensity 0.1 vs. 0.2).
> > |                        | 5%    | 10%   | 15%   | 20%   | 25%   | 30%   | 35%   | 40%   |
> > |------------------------|-------|-------|-------|-------|-------|-------|-------|-------|
> > | Ours                  | **0.909** | **0.943** | **0.995** | 1.043 | 1.075 | 1.127 | 1.163 | 1.222 |
> > | Hawkes (ground truth)           | 0.999 | 1.015 | 1.029 | **1.022** | **1.013** | **1.009** | **1.004** | **1.021** |
> > | Hawkes (misspecified) | 1.085 | 1.140 | 1.200 | 1.236 | 1.270 | 1.312 | 1.379 | 1.442 |
> >
> > Since forecasting STPPs without access to the underlying process is inherently challenging, especially for longer horizons, our model doesn't fully match the forecasting distribution for longer periods but achieves and even surpasses the NLL of the generative process for shorter forecasts.
> >
> > #### Log-likelihood
> > We agree that there are different approaches to computing the log-likelihood for (S)TPP models (hence our wording 'typically parameterized through the conditional intensity'), such as the normalized conditional intensity $p(x|H_{x_i})$ used in the two diffusion papers.
> > As noted in our previous response, traditional (S)TPP likelihood evaluation is inherently autoregressive, assessing only how well models predict the next event given its ground-truth history with known failure modes and providing minimal insights for real-world applications.
> > Since the contribution of our paper is to generalize beyond ordered point sets by generating all points in parallel, we cannot evaluate this autoregressively factorized likelihood.
> > Estimating this likelihood as proposed requires $points$ x $n_{samples}$ samples from our model per test set instance, with batches of 2000 samples processed in 2–3 seconds.
> > Thus, sampling a reasonable number of forecasts to estimate the 3D conditional density for one dataset and seed would require multiple days, making this estimate computationally prohibitive.
> >
> > Ultimately, we agree with the reviewer that likelihood computation is not a main flaw of our method, but we believe it highlights an important distinction from common (S)TPP models that warrants discussion.
> > Thus, for a camera-ready version, we will extend our discussion of this matter briefly presented in the conclusion (see lines 531-536).
> >
> > #### Release of code/implementation?
> > We will release the code with reproducible configurations of all experiments on GitHub upon acceptance.
> >
> > [1] Takahiro Omi, Naonori Ueda, and Kazuyuki Aihara. Fully neural network based model for general temporal point processes. In Advances in Neural Information Processing Systems, 2019.

---

### Meta-Review · Area_Chair_myKo · 2024-12-21

**Metareview:**

This paper proposes an interesting diffusion-based method for modeling and sampling point sets of point process distributions. For spatio-temporal point processes (STPP) the dominant approach is to apply autoregressive models that move points across time. This work compares to those methods, showing that they can sample point sets much more efficiently by parallelizing across time. Expert reviewers felt that the work was interesting, taking a new approach than prior work, and showing some clear advantages experimentally. We would be happy for this paper to be presented at ICLR. The authors already updated their appendix to address questions raised by the reviewers. We would encourage them to continue refining the work before the final version of the paper, and adding further experiments that could not be run due to time limitations.

**Additional Comments On Reviewer Discussion:**

There was a productive reviewer discussion, mainly between the authors and the most expert reviewer, Reviewer XRB4. The authors added several additional experiments to the paper and I think the discussion will also positively impact the presentation of the paper.

---

### Decision · Program_Chairs · 2025-01-22

Accept (Poster)